# Functional inhibition of acid sphingomyelinase disrupts infection by intracellular bacterial pathogens

Chelsea L Cockburn[1], Ryan S Green[1], Sheela R Damle[1], Rebecca K Martin[1], Naomi N Ghahrai[1], Punsiri M Colonne[2], Marissa S Fullerton[2], Daniel H Conrad[1], Charles E Chalfant[3], Daniel E Voth[2], Elizabeth A Rucks[4], Stacey D Gilk[5], Jason A Carlyon[1]

Intracellular bacteria that live in host cell–derived vacuoles are significant causes of human disease. Parasitism of low-density lipoprotein (LDL) cholesterol is essential for many vacuole-adapted bacteria. Acid sphingomyelinase (ASM) influences LDL cholesterol egress from the lysosome. Using functional inhibitors of ASM (FIASMAs), we show that ASM activity is key for infection cycles of vacuole-adapted bacteria that target cholesterol trafficking—*Anaplasma phagocytophilum*, *Coxiella burnetii*, *Chlamydia trachomatis*, and *Chlamydia pneumoniae*. Vacuole maturation, replication, and infectious progeny generation by *A. phagocytophilum*, which exclusively hijacks LDL cholesterol, are halted and *C. burnetii*, for which lysosomal cholesterol accumulation is bactericidal, is killed by FIASMAs. Infection cycles of Chlamydiae, which hijack LDL cholesterol and other lipid sources, are suppressed but less so than *A. phagocytophilum* or *C. burnetii*. *A. phagocytophilum* fails to productively infect $ASM^{-/-}$ or FIASMA-treated mice. These findings establish the importance of ASM for infection by intracellular bacteria and identify FIASMAs as potential host-directed therapies for diseases caused by pathogens that manipulate LDL cholesterol.

## Introduction

Intracellular bacteria that reside exclusively within host cell–derived vacuoles are major causes of disease in terms of both incidence and severity. If left untreated, the resulting infections can be severe, even fatal, or can become chronic and lead to extended periods of debilitation. Although certain antibiotics can effectively treat many of these diseases, bacterial resistance has been reported and allergy can occur (Jones et al, 1990; Lefevre et al, 1997; Somani et al, 2000; Spyridaki et al, 2002; Sandoz & Rockey, 2010;

Rouli et al, 2012), signifying the need for effective alternative therapeutics.

Parasitism of lipids, particularly cholesterol, is essential for intracellular bacterial pathogen infectivity [reviewed in Samanta et al (2017); Walpole et al (2018)]. Cholesterol is a major lipid component of eukaryotic membranes that influences membrane rigidity and is involved in diverse cellular processes including signal transduction, gene transcription, protein function and degradation, endocytic and Golgi trafficking, and intra-organelle membrane contact site formation. In mammalian cells, whereas cholesterol can be synthesized de novo in the endoplasmic reticulum, most is acquired exogenously via the low-density lipoprotein (LDL) receptor. After LDL uptake, esterified cholesterol is trafficked by the endocytic route to lysosomes, where it is hydrolyzed to unesterified free cholesterol molecules that are delivered to the plasma membrane, *trans*-Golgi network (TGN), endoplasmic reticulum, and ultimately throughout the cell (Urano et al, 2008; Samanta et al, 2017; Walpole et al, 2018). Lysosomes therefore play an essential role in intracellular cholesterol homeostasis (Kuzu et al, 2017).

Inhibition of lysosomal cholesterol efflux occurs in lipid storage disorders, such as Niemann–Pick disease (Brady et al, 1966). The type A and B forms of this condition result from loss of function mutations in acid sphingomyelinase (ASM), a lysosomal enzyme that hydrolyzes sphingomyelin to yield phosphorylcholine and ceramide (Vanier, 2013). ASM deficiency leads to sphingomyelin accumulation in lysosomes, which, in turn, blocks LDL-derived cholesterol efflux (Lloyd-Evans et al, 2008). Niemann–Pick disease severity correlates with decreased ASM activity (Schuchman & Miranda, 1997). Conversely, ASM activation has also been linked to the development of multiple human diseases [reviewed in Schuchman (2010); Kornhuber et al (2015)], and studies using cells from Niemann–Pick patients or ASM knockout mice indicate that ASM deficiency might also have beneficial consequences [reviewed in Kornhuber et al (2010)]. Indeed, functional inhibitors of ASM (FIASMAs) have emerged as promising drugs with broad therapeutic

[1]Department of Microbiology and Immunology, Virginia Commonwealth University Medical Center, School of Medicine, Richmond, VA, USA   [2]Department of Microbiology and Immunology, University of Arkansas for Medical Sciences, Little Rock, AR, USA   [3]Department of Cell Biology, Microbiology, and Molecular Biology, University of South Florida, Tampa, FL, USA   [4]Department of Pathology and Microbiology, University of Nebraska Medical Center, Omaha, NE, USA   [5]Department of Microbiology and Immunology, Indiana University School of Medicine, Indianapolis, IN, USA

Correspondence: jason.carlyon@vcuhealth.org

potential (Kornhuber et al, 2010; Kuzu et al, 2017). FIASMAs are lysosomotropic compounds that indirectly inactivate ASM by promoting its detachment from the inner lysosomal membrane, rendering it susceptible to proteolysis. FIASMAs are active in cell culture models and in vivo at concentrations that are therapeutically achieved during pharmacotherapy in humans. Many FIASMAs are FDA approved for clinical use in humans (Kornhuber et al, 2010).

Some studies have investigated ASM's relevance to microbial infections and FIASMAs' potential therapeutic benefit in this context. For instance, FIASMA treatment protects mice from superoxide-mediated lung edema associated with *Staphylococcus aureus* infection and prevents lethal *S. aureus* sepsis when administered together with antibiotics (Peng et al, 2015). Also, paradoxically, whereas ASM-mediated phagosome maturation is important for controlling mycobacterial infection, ASM-dependent cell–cell fusion can provide an innate immunoescape niche for mycobacterial replication (Utermohlen et al, 2008; Vazquez et al, 2016; Wu et al, 2018). Given that multiple intracellular bacterial pathogens hijack LDL cholesterol trafficking and storage pathways for growth and/or survival [reviewed in Samanta et al (2017); Walpole et al (2018)], FIASMAs could represent novel, non-antibiotic means for treating the diseases that these bacteria cause. Yet, their potential in this capacity and the importance of ASM in intracellular bacterial infections that involve cholesterol parasitism have gone largely unexplored.

Here, we demonstrate that ASM activity is essential for optimal infection cycle progression of four obligate intracellular vacuole-adapted bacterial pathogens that target host cholesterol trafficking pathways: *Anaplasma phagocytophilum* (Xiong et al, 2009; Xiong & Rikihisa, 2012), *Coxiella burnetii* (Howe & Heinzen, 2006; Mulye et al, 2018), *Chlamydia trachomatis* (Carabeo et al, 2003; Beatty, 2006, 2008; Kumar et al, 2006; Cocchiaro et al, 2008; Cox et al, 2012), and *Chlamydia pneumoniae* (Liu et al, 2010). The degree of FIASMA-mediated inhibition correlates with pathogen dependency on LDL cholesterol. ASM-deficient mice are resistant to *A. phagocytophilum* infection and FIASMA administration postinfection prevents the bacterium from productively infecting wild-type (WT) mice. Overall, this study establishes the importance of ASM to infection by multiple intracellular bacteria and distinguishes FIASMAs as potential therapeutics for diseases caused by pathogens whose growth is influenced by LDL cholesterol.

## Results

### Functional inhibition of host cell ASM reduces the *A. phagocytophilum* load

*A. phagocytophilum* infects neutrophils to cause the emerging disease human granulocytic anaplasmosis, which presents as an acute nonspecific febrile illness that can progress to severe complications or death in immunocompromised patients, the elderly, and in the absence of antibiotic intervention (Ismail & McBride, 2017). *A. phagocytophilum* lacks genes required for lipid A biosynthesis and most peptidoglycan synthesis genes (Lin & Rikihisa, 2003; Dunning Hotopp et al, 2006). The bacterium incorporates cholesterol into its fragile cell envelope and requires the lipid for intracellular replication, but is devoid of genes encoding cholesterol biosynthesis or modification enzymes and must parasitize the sterol from host cells (Lin & Rikihisa, 2003). *A. phagocytophilum* obtains cholesterol exclusively by hijacking the Niemann–Pick type C protein 1 (NPC1) pathway that mediates lysosomal cholesterol efflux (Xiong et al, 2009; Xiong & Rikihisa, 2012), which makes it an ideal organism for evaluating the efficacy of FIASMAs for inhibiting infection by an LDL cholesterol–dependent pathogen.

Promyelocytic HL-60 and RF/6A endothelial cells are established models for examining *A. phagocytophilum*–host cell interactions (Klein et al, 1997; Munderloh et al, 2004; Truchan et al, 2016c). Desipramine is an FDA-approved tricyclic antidepressant that functionally inhibits ASM (Kornhuber et al, 2010). To determine if pharmacologic inhibition of ASM inhibits *A. phagocytophilum* infection, desipramine-treated HL-60 and RF/6A cells were incubated with *A. phagocytophilum*. PCR analyses at 24, 48, and 72 h revealed that, although bacterial DNA levels increased throughout the time course in control cells, they were pronouncedly reduced and did not increase in desipramine-treated cells in a dose-dependent manner (Fig 1A–C). Desipramine also halted *A. phagocytophilum* infection in human neutrophils (Fig 1D). This experiment was only carried out for 32 h to allow completion of one bacterial infection cycle because, although *A. phagocytophilum* extends the 12-h half-life of neutrophils (Alberdi et al, 2016), cell death was observed after 32 h. Desipramine prevented an increase in *A. phagocytophilum* load when administered to HL-60 cells at 24 h postinfection, thereby indicating its ability to inhibit active infection (Fig 1E). However, desipramine treatment had no effect on bacterial binding to host cells (Fig 1F). Although many bacterial sphingomyelinases function as virulence factors (Flores-Diaz et al, 2016), none are encoded by the annotated *A. phagocytophilum* genome (Dunning Hotopp et al, 2006). Nonetheless, to verify that the inhibitory effect of desipramine on *A. phagocytophilum* infection in host cells was not due to the drug directly acting on the bacterium, host cell–free *A. phagocytophilum* organisms were exposed to the drug or vehicle before incubation with HL-60 cells. The bacterial DNA load at 24 h postinfection was equivalent between host cells that had been pretreated with desipramine or that had not been treated (Fig 1G).

In addition to functionally inhibiting ASM, desipramine promotes degradation of acid ceramidase, which acts downstream of ASM to hydrolyze ceramide to sphingosine (Elojeimy et al, 2006; Kornhuber et al, 2010). Therefore, to confirm that desipramine's deleterious effect on *A. phagocytophilum* was specific to its action on ASM, HL-60 cells were pretreated with CA-074 Me, which blocks desipramine's effect on acid ceramidase (Elojeimy et al, 2006), before successive incubations with desipramine and *A. phagocytophilum*. CA-074 Me failed to abrogate the desipramine-mediated arrest of *A. phagocytophilum* infection (Fig 1H). Amitriptyline and nortriptyline, two other FDA-approved tricyclic antidepressants and confirmed FIASMAs (Kornhuber et al, 2010), also suppress *A. phagocytophilum* infection (Fig 1 I and J). Collectively, these data indicate that pharmacologic inhibition of host cell–ASM activity prevents an increase in *A. phagocytophilum* load at a post-bacterial adhesion step and in a dose-dependent manner.

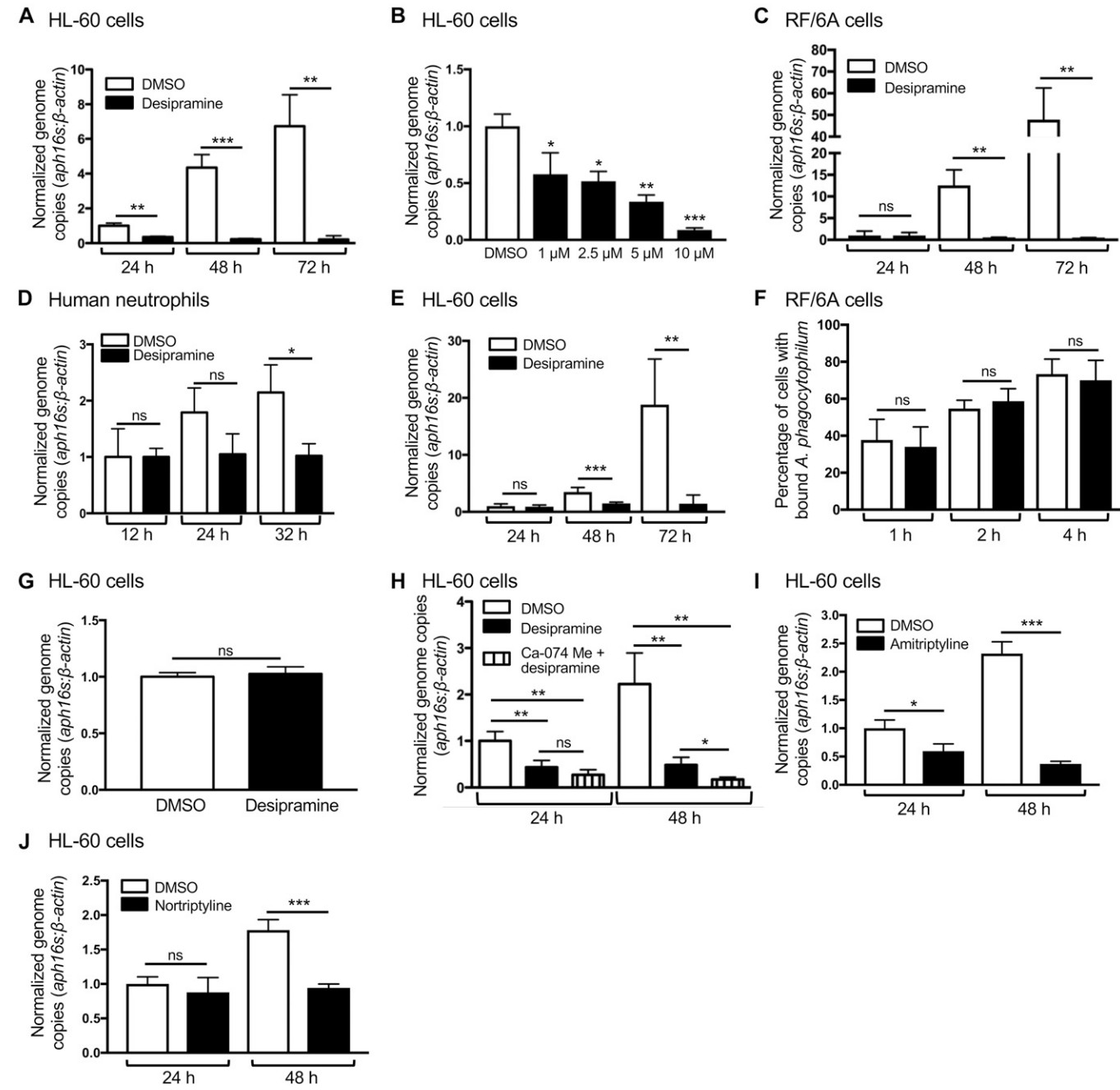

**Figure 1. FIASMAs inhibit *A. phagocytophilum* infection at a post bacterial invasion step through specific targeting of host cell ASM.**
**(A–E, I, J)** FIASMA treatment reduces the *A. phagocytophilum* DNA load in host cells. HL-60 cells (A, B, I, J), RF/6A cells (C), or human peripheral blood neutrophils (D) were treated with 10 μM (unless otherwise noted) desipramine (A, B), amitriptyline (I), nortriptyline (J), or DMSO vehicle control followed by incubation with *A. phagocytophilum* organisms. Total DNA isolated at the indicated time points was analyzed by qPCR. Relative levels of the *A. phagocytophilum* 16S rRNA (*aph16s*) gene were normalized to the relative levels of *β-actin* using the $2^{-\Delta\Delta CT}$ method. **(E)** Desipramine was added to *A. phagocytophilum*–infected cells beginning at 24 h followed by qPCR analysis. **(F)** Desipramine has no effect on *A. phagocytophilum* binding to host cells. RF/6A cells were exposed to desipramine or DMSO followed by incubation with *A. phagocytophilum*. At 1, 2, and 4 h, the cells were fixed, immunolabeled with antibody against the *A. phagocytophilum* surface protein, P44, and examined by immunofluorescence microscopy to determine the percentages of cells having bound *A. phagocytophilum* organisms. **(G)** Desipramine treatment of *A. phagocytophilum* does not alter infection of host cells. Host cell–free *A. phagocytophilum* bacteria were exposed to desipramine or DMSO followed by incubation with untreated HL-60 cells. At 24 h, the bacterial load was determined using qPCR. **(H)** The inhibitory effect of desipramine on *A. phagocytophilum* infection is due to its action on ASM, not acid ceramidase. HL-60 cells were treated with CA-074 Me or not followed by treatment with desipramine or vehicle control. The cells were then infected with *A. phagocytophilum*. At 24 and 48 h, the bacterial load was measured using qPCR. Error bars indicate SD. *t* test was used to test for a significant difference among pairs. One-way ANOVA with Tukey's post hoc test was used to test for a significant difference among groups. Statistically significant (*$P < 0.05$; **$P < 0.01$; ***$P < 0.001$) values are indicated. ns, not significant. Data shown in (A) are representative of three experiments conducted in triplicate with similar results. Data in (B, D, F–J) are representative of two experiments conducted in triplicate with similar results. Data in (C) are representative of five experiments conducted in triplicate with similar results. Data in (E) are representative of seven experiments conducted in triplicate with similar results.

### ASM inhibition halts *A. phagocytophilum* vacuole maturation and expansion

Because FIASMAs arrest *A. phagocytophilum* infection at a post-bacterial binding step, we sought to identify the specific life cycle stage(s) affected. The *A. phagocytophilum* biphasic developmental cycle initiates when an infectious dense-core (DC) organism bound at the host cell surface enters via receptor-mediated endocytosis (Mott et al, 1999; Troese & Carlyon, 2009). Within the first four h, the DC transitions to the noninfectious, replicative reticulate cell (RC) form and actively remodels its vacuole such that it avoids lysosomal fusion and becomes wrapped in vimentin intermediate filaments (Webster et al, 1998; Troese & Carlyon, 2009; Truchan et al, 2016b). Over the next 24 h, RCs divide within the inclusion as they expand in size, interface with organelles, and are decorated with secreted bacterial effector proteins (Troese & Carlyon, 2009; Huang et al, 2010a, 2010b; Niu et al, 2012; Truchan et al, 2013, 2016b). RC to DC conversion occurs between 24 and 32 h followed by the release of infectious DC progeny between 28 and 36 h (Troese & Carlyon, 2009).

We first examined whether functionally inhibiting ASM impedes *A. phagocytophilum* vacuole (ApV) maturation. Bacterial inclusions in infected RF/6A cells were assessed using confocal microscopy for the accumulation of vimentin, which is recruited early and remains irreversibly associated with the ApV for the entire infection cycle (Truchan et al, 2016b), and for the presence of APH0032, an *A. phagocytophilum* effector that is expressed and localizes to the ApV membrane during late-stage infection, between 20 and 32 h (Huang et al, 2010a; Oki et al, 2016). RF/6A cells were selected for this purpose because they are large and flat, enabling optimal imaging of the ApV (Munderloh et al, 2004; Sukumaran et al, 2011; Truchan et al, 2016c). In control cells, APH0032 was detected on an increasing percentage of ApVs at 20, 24, 28, and 32 h (Fig 2 A and B), suggesting that ApV maturation progressed normally. In desipramine-treated cells, however, pronouncedly fewer APH0032-positive ApVs were detected. Vimentin is associated with ApVs observed under both conditions (Fig 2A), indicating that this early ApV biogenesis event is not dependent on ASM activity. Indeed, a separate experiment verified that vimentin filaments wrap 100% of ApVs in RF/6A cells irrespective of whether they are treated with DMSO or desipramine (Fig S1A and B). Consistent with desipramine being a reversible ASM inhibitor (Kornhuber et al, 2010), after the removal of the drug at 20 h, the numbers of APH0032-positive ApVs began to significantly increase by 28 h compared with cultures that contained desipramine (Fig 2B). The inverse phenomenon was observed when desipramine was first added at 20 h postinfection, as the percentage of APH0032-positive ApVs did not increase and was significantly less than that for control cells beginning at 28 h (Fig 2C). ApV area increased throughout the time course in control cells, but not in desipramine-treated cells (Fig 2D). Similar to APH0032 ApV membrane localization (Fig 2A–C), desipramine's effect on ApV size was bacteriostatic, as removal or addition at 20 h enabled restoration or stagnation of ApV expansion by 24 h, respectively (Fig 2E and F). Overall, these data demonstrate that functional inhibition of ASM inhibits late-stage expansion and maturation of the ApV in a reversible manner.

### Desipramine inhibits *A. phagocytophilum* infectious progeny generation

ApV maturation, in terms of APH0032 localization to the ApV membrane, precedes *A. phagocytophilum* RC to DC conversion (Truchan et al, 2016c). FIASMA treatment reduces bacterial load and inhibits ApV maturation and expansion. Accordingly, we rationalized that desipramine impedes the production of infectious DC progeny. To test this hypothesis, HL-60 cells were exposed to desipramine or vehicle followed by infection with *A. phagocytophilum*. qRT-PCR was performed using the total RNA isolated at 24, 28, and 32 h, the period during which RC to DC conversion occurs (Troese & Carlyon, 2009), to measure transcript levels of *aph1235*, a DC-specific protein that contributes to *A. phagocytophilum* infectivity (Troese et al, 2011; Mastronunzio et al, 2012). An RC-unique marker has yet to be identified. A similar experiment was performed in parallel in which RF/6A cells treated with desipramine and incubated with *A. phagocytophilum* were screened for ApVs harboring APH1235-positive bacteria. As previously observed (Troese et al, 2011; Truchan et al, 2016c), both *aph1235* expression and the number of ApVs harboring APH1235-positive bacteria increased throughout the time course for control cells (Fig 3A–C). In desipramine-treated cells, *aph1235* levels and ApVs containing APH1235-immunolabeled bacteria were pronouncedly reduced. To resolve whether the overall reduction in *A. phagocytophilum* load in desipramine-treated cells is specifically due to impairment of conversion to the infectious form and not due to a blockade in release of infectious progeny, infected desipramine-treated or control RF/6A cells were mechanically disrupted at 48 h. Released bacteria were incubated with naïve cells followed by immunofluorescence microscopic examination of infection 24 h later. The percentage of infected cells after incubation with *A. phagocytophilum* organisms recovered from desipramine-treated cells was eightfold lower than cells incubated with bacteria recovered from control cells (Fig 3D). These data confirm that functional inhibition of ASM inhibits *A. phagocytophilum* conversion to the infectious form.

### Desipramine halts the *A. phagocytophilum* infection cycle by inhibiting NPC1-mediated trafficking of cholesterol to the ApV

Two lysosomal proteins, NPC1 and Niemann–Pick type C protein 2 (NPC2), cooperate to export cholesterol from lysosomes. NPC2 extracts and transfers cholesterol from the lysosomal internal membrane to NPC1, which aids in moving cholesterol from the limiting membrane to subcellular destinations via vesicular transport (Kuzu et al, 2017). Lysosomal accumulation of sphingomyelin resulting from deficiency or inhibition of ASM interferes with the ability of NPC2 to transfer cholesterol to NPC1, which, in turn, prevents NPC1-mediated transport of LDL cholesterol and its accumulation within lysosomes (Abdul-Hammed et al, 2010; Oninla et al, 2014; Kuzu et al, 2017). Given that *A. phagocytophilum* is dependent on the NPC1 pathway (Xiong & Rikihisa, 2012), we directly assessed whether desipramine inhibits NPC1-mediated cholesterol trafficking to the ApV. Immunolabeled NPC1 was detected in close proximity to the ApV membrane and within the vacuole lumen associated with *A. phagocytophilum* organisms in vehicle

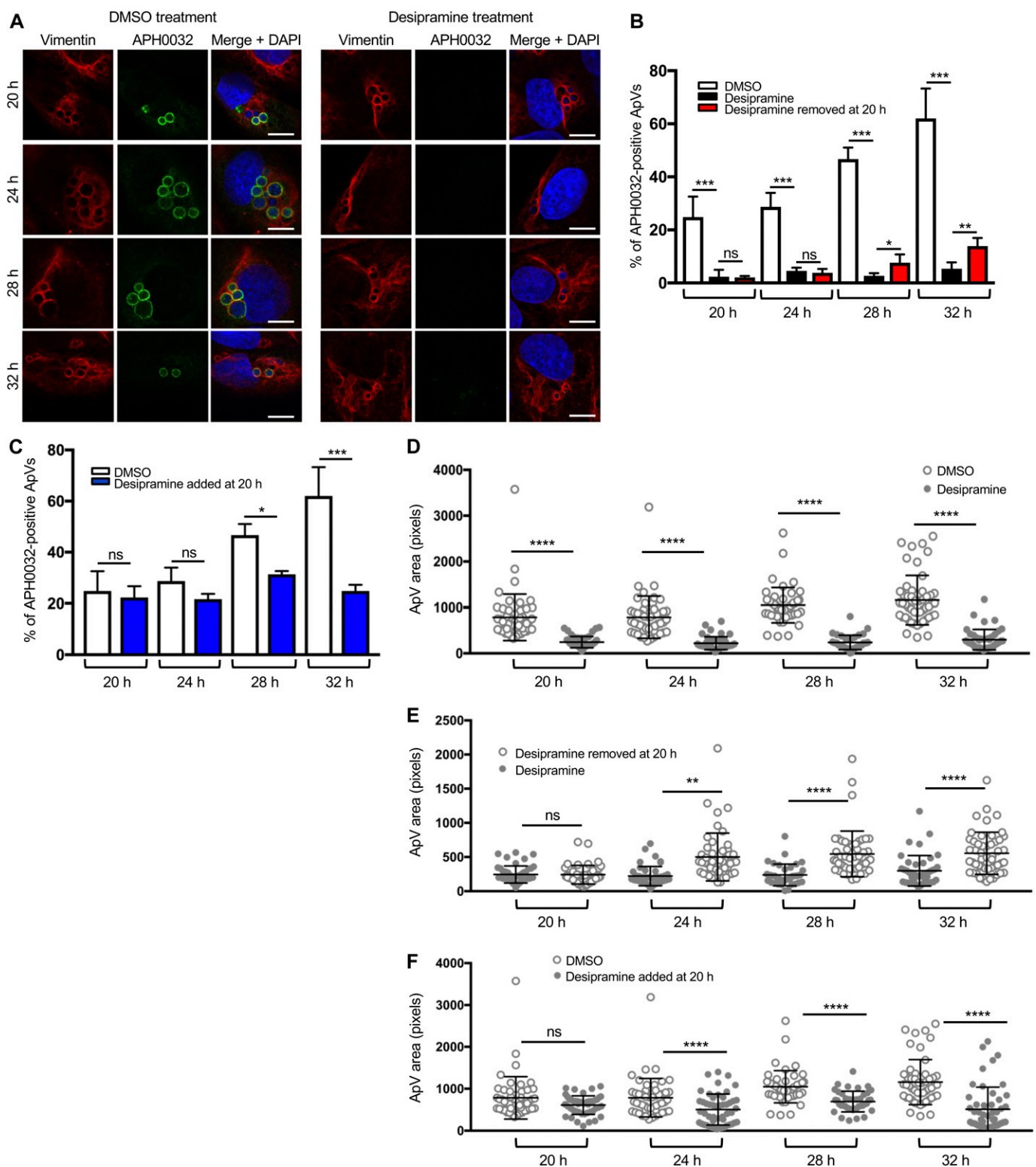

**Figure 2. Functional inhibition of ASM halts ApV maturation and expansion.**
**(A, B, D, E)** Desipramine was added to RF/6A cells before infection with *A. phagocytophilum* and treatment was either maintained throughout the time course (A, B, D) or removed at 20 h (B, E). **(C, F)** Desipramine was added to *A. phagocytophilum*–infected RF/6A cells beginning at 20 h (C, F). DMSO served as vehicle control. At 20, 24, 28, and 32 h, the cells were fixed and examined by confocal microscopy for ApV maturation (A–C) and expansion (D–F). **(A–C)** Desipramine reversibly inhibits APH0032 expression and localization to the ApV. *A. phagocytophilum*–infected RF/6A cells were screened with antibodies targeting vimentin and APH0032 to demarcate and assess maturation of the ApV, respectively. DAPI was used to stain host cell nuclei and bacterial DNA. **(A)** Representative confocal micrographs of desipramine or DMSO-treated cells at the

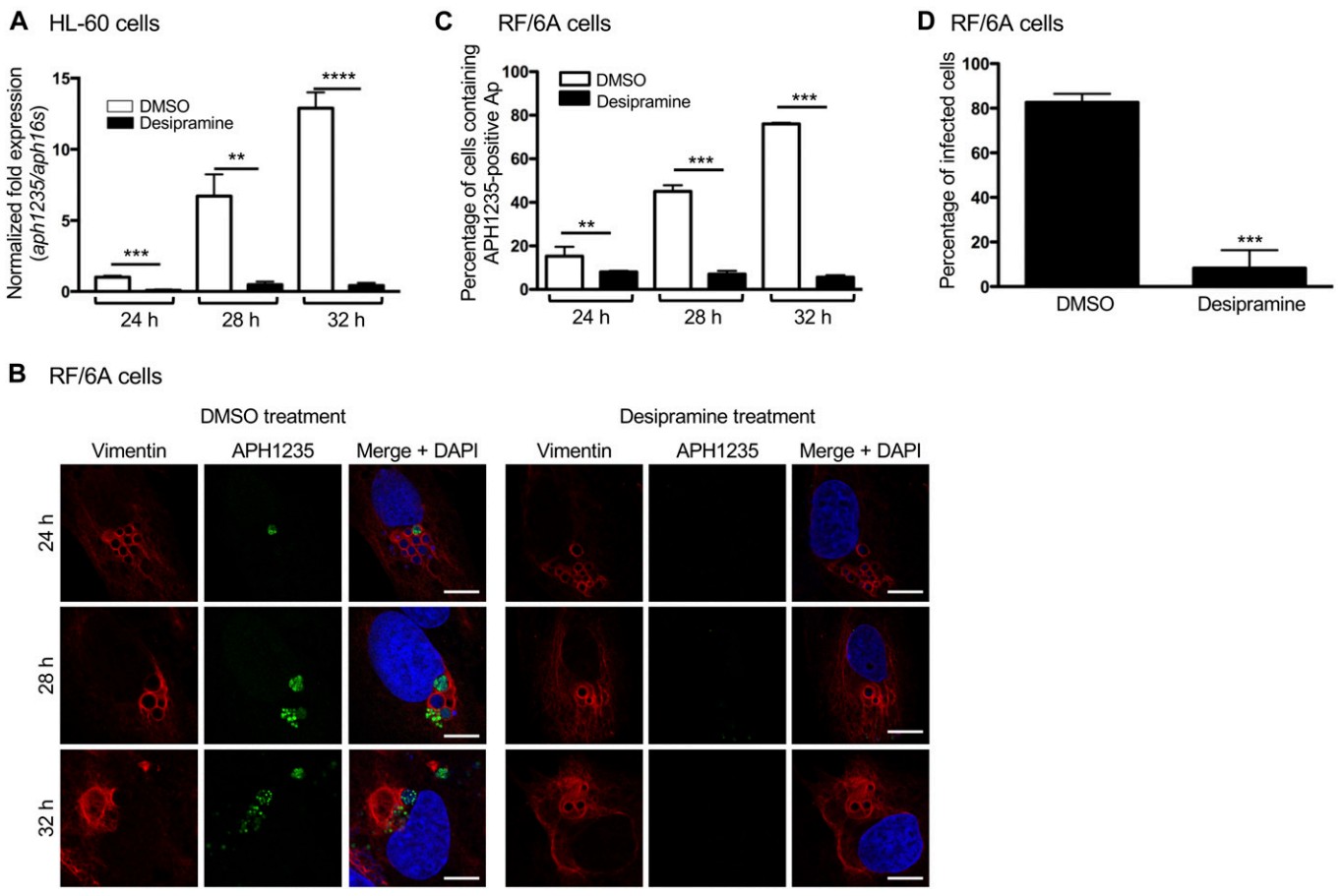

**Figure 3. Desipramine inhibits *A. phagocytophilum* conversion to the infectious form.**
**(A)** Desipramine inhibits *aph1235* transcription. Desipramine-treated HL-60 cells were infected with *A. phagocytophilum*. Total RNA isolated at 24, 28, and 32 h was subjected to qRT-PCR. The $2^{-\Delta\Delta CT}$ method was used to determine the relative *aph1235* expression level normalized to that of *A. phagocytophilum* 16S rRNA. **(B, C)** Desipramine inhibits APH1235 protein expression. Desipramine-treated RF/6A cells were infected with *A. phagocytophilum*. At 24, 28, and 32 h, the cells were fixed, immunolabeled with APH1235 and vimentin antibodies, stained with DAPI, and visualized using confocal microscopy. **(B)** Representative confocal micrographs. Scale bar = 10 μM. **(C)** Percentage of APH1235-positive ApVs determined by counting 100 cells for each of triplicate samples per time point. **(D)** Desipramine inhibits *A. phagocytophilum*–infectious progeny production. RF/6A cells were treated with desipramine or DMSO followed by infection with *A. phagocytophilum*. At 48 h, the cells were mechanically disrupted followed by isolation and subsequent incubation of host cell–free bacteria with naïve untreated cells. At 24 h, the recipient cells were fixed and examined by immunofluorescence microscopy to determine the percentage that had become infected. Error bars indicate SD. *t* test was used to test for a significant difference among pairs. Statistically significant (\*\**P* < 0.01; \*\*\**P* < 0.001; \*\*\*\**P* < 0.0001) values are indicated. Data shown in (A–C) are representative of three experiments conducted in triplicate that yielded similar results. Data shown in (D) are representative of two experiments conducted in triplicate with similar results.

control–treated RF/6A cells, but not in desipramine-treated cells (Fig 4A and B). *A. phagocytophilum* intracellular parasitism also involves hijacking Beclin-1 to induce autophagosome formation and rerouting ceramide-rich TGN46-positive *trans*-Golgi–derived vesicles to the ApV (Niu et al, 2012; Truchan et al, 2016c). Yet, desipramine had no effect on either of these phenomena (Fig S1C and D). Live cell imaging revealed that desipramine drastically reduced the percentage of ApVs with which BODIPY cholesterol–positive, but not BODIPY ceramide–positive vesicles associated (Fig 4C–E). Consistent with blocking cholesterol egress from lysosomes, desipramine induced the formation of numerous large BODIPY cholesterol–filled vesicles (Fig 4C) that were confirmed to be

LysoTracker Red positive (Fig S1E). Thus, functional inhibition of ASM blocks *A. phagocytophilum* infection cycle progression by specifically interfering with bacterial acquisition of LDL cholesterol via the NPC1 pathway.

## ASM is essential for *A. phagocytophilum* to productively infect mice

Because ASM activity is critical for *A. phagocytophilum* infection to progress in tissue culture cells, the relevance of ASM to infection in vivo was determined. Groups of WT or ASM$^{-/-}$ mice were inoculated with DC organisms followed by qPCR analyses of bacterial DNA load

indicated postinfection time points. **(B, C)** Percentages of APH0032-positive ApVs determined for 100 cells for each of three biological replicates per condition. **(D–F)** Desipramine reversibly inhibits ApV expansion. The mean ApV pixel area was determined for 50 ApVs per time point per condition. Error bars indicate SD. *t* test was used to test for a significant difference among pairs. Statistically significant (\**P* < 0.05; \*\**P* < 0.01; \*\*\**P* < 0.001; \*\*\*\**P* < 0.0001) values are indicated. ns, not significant. Data shown are representative of three experiments conducted in triplicate with similar results. Scale bar = 10 μm.

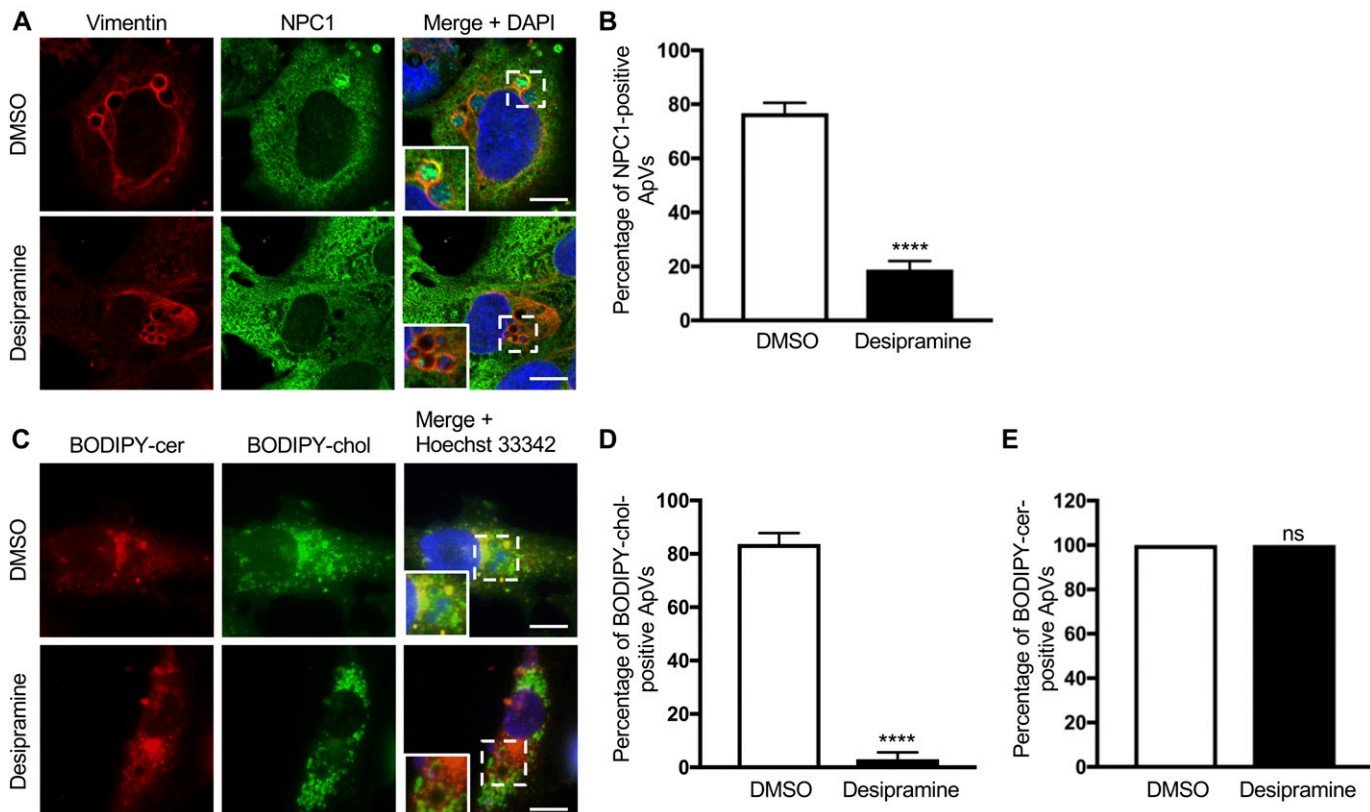

**Figure 4. Desipramine alters NPC1-mediated cholesterol trafficking to the ApV.**
Desipramine treatment inhibits NPC1 localization to the ApV. Desipramine- or DMSO-treated RF/6A cells were infected *A. phagocytophilum*. **(A–E)** At 24 h, the cells were either fixed, immunolabeled with vimentin and NPC1 antibodies, stained with DAPI, and examined by confocal microscopy (A, B); or incubated with BODIPY ceramide (cer) or BODIPY cholesterol (chol), stained with Hoechst 33342, and visualized by live cell imaging (C–E). **(A)** Representative confocal micrographs of infected cells immunolabeled for vimentin and NPC1. Regions that are demarcated by hatched-line boxes are magnified in the inset panels. **(B)** Percentage of vimentin-positive ApVs to which NPC1 immunosignal localizes in DMSO- and desipramine-treated cells determined by counting 100 cells for each of triplicate samples per condition. **(C)** Representative live cell images of infected cells incubated with BODIPY-cer and BODIPY-chol. **(D, E)** Percentages of ApVs to which BODIPY-cer–positive (D) or BODIPY-chol–positive (E) vesicles localize. Error bars indicate SD. *t* test was used to test for a significant difference among pairs. Statistically significant (****$P < 0.001$) values are indicated. ns, not significant. Data shown are representative of three experiments conducted in triplicate that yielded similar results. Scale bar = 10 $\mu$m.

in the peripheral blood. The *A. phagocytophilum* load in WT mice peaked by day 16 and gradually subsided to an undetectable level by day 28 (Fig 5A). In ASM$^{-/-}$ mice, however, bacterial DNA was barely detectable at any time point. To assess desipramine's ability to reduce bacterial burden in an active infection, WT mice were inoculated with *A. phagocytophilum* followed by injection with either desipramine or vehicle on days 7 through 12 at a dosage confirmed to inhibit ASM activity in vivo and within the range approved for use in humans (Teichgraber et al, 2008; Hayasaka et al, 2015). Strikingly, although infection proceeded normally in vehicle-treated mice, the *A. phagocytophilum* burden was drastically reduced on days 8 and 12 and no longer detectable beginning on day 16 in desipramine-treated mice (Fig 5B). These data confirm that ASM is critical for and demonstrate the ability of desipramine to eliminate *A. phagocytophilum* infection in vivo.

### Desipramine-induced cholesterol accumulation in the *C. burnetii* parasitophorous vacuole is bactericidal

To investigate whether desipramine's efficacy in inhibiting infection could be extended to other intracellular bacteria that interface with

LDL cholesterol trafficking pathways, we next focused on *C. burnetii*, which causes Q fever. Whereas acute Q fever is typically self-limiting, chronic disease requires at least 18 mo of antibiotic therapy. *C. burnetii* first infects alveolar macrophages during natural infection, but can be found in a wide range of cell types. Inside the host cell, *C. burnetii* directs formation of a large, lysosome-like vacuole called the *Coxiella*-containing vacuole (CCV) (Kohler & Roy, 2015). Although cholesterol, including LDL-derived cholesterol, readily traffics to the CCV, *C. burnetii* is sensitive to cholesterol levels in the CCV membrane, with elevated cholesterol leading to increased acidification and bacterial degradation (Mulye et al, 2017). Furthermore, *C. burnetii* is sensitive to drugs that perturb host cholesterol homeostasis and grows poorly in NPC1-deficient cells (Czyz et al, 2014; Howe & Heinzen, 2006). Given the sensitivity of *C. burnetii* to cholesterol, we tested the effect of desipramine on *C. burnetii* growth in THP-1 macrophage-like cells. Desipramine-treated or untreated control cells were incubated with trans-genic *C. burnetii* constitutively expressing mCherry. Cells were cultivated in the continued presence of drug or vehicle and fluo-rescence levels measured daily as an indication of bacterial growth. Beginning on day 3, the first day on which there was any detectable

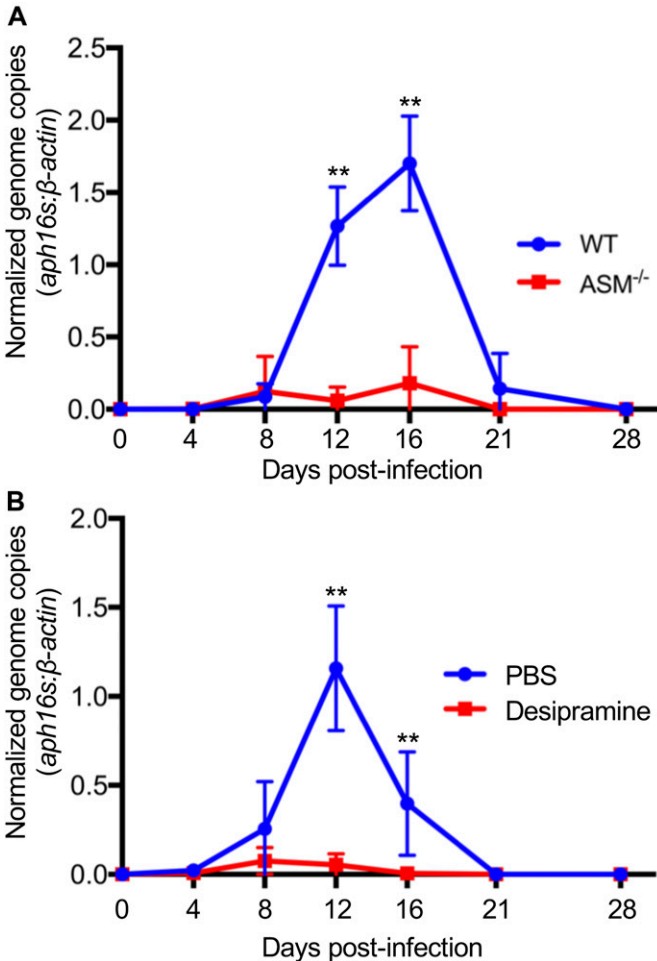

**Figure 5. ASM is essential for *A. phagocytophilum* to productively infect mice.** **(A)** *A. phagocytophilum* fails to productively infect ASM$^{-/-}$ mice. **(A)** ASMase$^{-/-}$ mice or WT mice were infected with *A. phagocytophilum* DC organisms. Peripheral blood drawn on days 4, 8, 12, 16, 21, and 28 d postinfection was analyzed by qPCR. Relative levels of the *A. phagocytophilum* 16S rRNA gene were normalized to those of *β-actin* using the 2$^{-\Delta\Delta CT}$ method. **(B)** Desipramine reduces the *A. phagocytophilum* DNA load in the peripheral blood when administered to infected mice. *A. phagocytophilum*–infected WT mice were administered desipramine or PBS on days 7 through 12 postinfection, and the bacterial DNA load in the peripheral blood was determined using qPCR. Error bars indicate SD. *t* test was used to test for a significant difference among pairs. Statistically significant (**P* < 0.01) values are indicated. Data shown in (A) are representative of eight experiments each of which were conducted with 5–7 mice per group. Data shown in (B) are representative of three experiments each of which were conducted with 5–7 mice per group.

increase in fluorescence signal, mCherry–*C. burnetii* proliferation was significantly inhibited in desipramine-treated cells (Fig 6A). Importantly, desipramine failed to alter mCherry–*C. burnetii* growth in axenic medium (Fig 6B), indicating that its inhibitory effect is restricted to the intracellular niche. To determine whether desipramine's action on the pathogen was bacteriostatic or bactericidal, *C. burnetii* was allowed to infect HeLa cells and MH-S murine alveolar macrophages treated with desipramine or DMSO. Cells were harvested every 48 h to determine the number of viable bacteria using a colony-forming unit (CFU) assay (Clemente et al, 2018). In control HeLa cells, a fourfold increase in bacterial load was observed, whereas the number of viable *C. burnetii* decreased over

time in desipramine-treated cells (Fig 6C). Comparable results were observed in MH-S macrophages (Fig 6D), suggesting that desipramine is bactericidal to *C. burnetii*.

Similar to *A. phagocytophilum*, *C. burnetii* undergoes a biphasic developmental cycle. The environmentally stable but non-replicative small-cell variant infects cells through phagocytosis, and the bacteria-containing phagosome matures through the default endocytic pathway to a phagolysosome. Over the next 24–48 h, bacteria reside in the tight-fitting phagolysosome and transition to the replicative large-cell variant. Around 48 h, the CCV expands through fusion with host endosomal compartments and auto-phagosomes, allowing bacterial replication (Kohler & Roy, 2015). To define the stage of the *C. burnetii* intracellular life cycle that is desipramine sensitive, we added the drug at 24-h intervals after infection and determined bacterial numbers after 6 d. *C. burnetii* was sensitive to desipramine only in the first 48 h after infection (Fig 6E), suggesting that inhibiting ASM affects an early stage of *C. burnetii* infection before CCV expansion. This result is further supported by the observation that CCVs in desipramine-treated cells did not expand, based on CCV size at 6 d postinfection (Fig 6F).

Previous experiments demonstrated that blocking cholesterol export from late endosomes/lysosomes with U18666A increased cholesterol in the CCV, leading to acidification and bacterial death (Mulye et al, 2017). To determine if functionally inhibiting ASM increases CCV cholesterol levels, we labeled infected cells with filipin, a fluorescent cholesterol-binding compound. As observed previously (Howe & Heinzen, 2006; Mulye et al, 2017), filipin labeled the CCV membrane in control cells (Fig 6G), indicating the CCV contains cholesterol or other sterols. However, upon desipramine treatment, the intensity of filipin labeling in the CCV increased, confirming that ASM inhibition elevates CCV cholesterol levels. Together, these data suggest that blocking ASM activity kills *C. burnetii* by increasing CCV cholesterol levels.

### Chlamydiae do not demonstrate the same degree of sensitivity to desipramine as *A. phagocytophilum* and *C. burnetii*

Given that the exquisite sensitivities of *A. phagocytophilum* and *C. burnetti* to desipramine are linked to interactions with LDL cholesterol in the NPC1 pathway, we examined whether the drug could effectively target obligate intracellular bacteria whose growth is not exclusively dependent on LDL cholesterol. *C. trachomatis*, a leading cause of sexually transmitted disease and infectious blindness, obtains LDL-derived and de novo–synthesized cholesterol by intercepting exocytic traffic from the Golgi (Carabeo et al, 2003). *C. trachomatis* also recruits cholesterol-rich multivesicular bodies (MVBs) and high-density lipoprotein biogenesis proteins involved in cholesterol efflux to its inclusion for growth (Beatty, 2006, 2008; Cox et al, 2012). To determine if desipramine alters *C. trachomatis* infection, desipramine-treated and control HeLa cells were infected with *C. trachomatis* serovar L2 for 28–30 h. Infected cells were either fixed to verify equal infection across conditions or lysed. The lysates were serially diluted and plated onto fresh monolayers, which were assessed for inclusion-forming units at 24 h post-infection. Desipramine treatment throughout the chlamydial developmental cycle resulted in biologically irrelevant, albeit

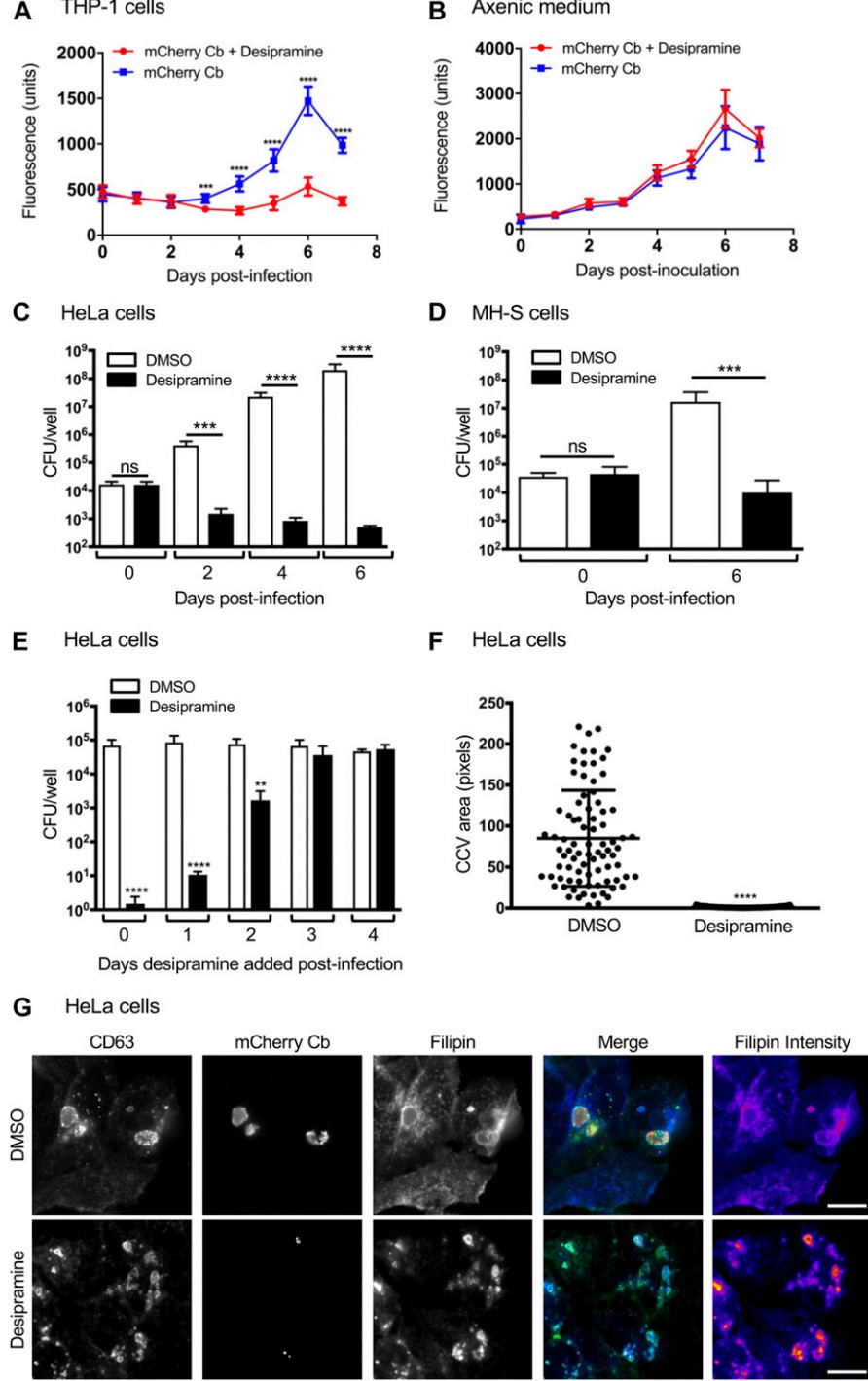

**Figure 6. Desipramine-induced cholesterol accumulation in the *C. burnetii* vacuole is bactericidal.**
**(A, B)** mCherry-*C. burnetii* (Cb)–infected THP-1 macrophage-like cells (A) or mCherry-*C. burnetii* grown in axenic medium (B) were treated with desipramine or not treated with. The bacterial load was measured as relative fluorescent units. **(C–E)** *C. burnetii* was added to HeLa cells (C, E) or MH-S cells (D) that had been pretreated with desipramine or DMSO, or *C. burnetii*–infected cells were treated at the indicated days postinfection (E). Bacterial load was measured using a CFU assay. **(F)** CCV area was determined for desipramine and DMSO-treated *C. burnetii*–infected HeLa cells. **(G)** HeLa cells that had been treated with desipramine and infected with mCherry-*C. burnetii* were labeled with filipin and CD63 antibody. Error bars indicate SD. *t* test was used to test for a significant difference among pairs. Statistically significant (\*\**P* < 0.01; \*\*\**P* < 0.001; \*\*\*\**P* < 0.0001) values are indicated. ns, not significant. Data in panels A and B are representative of three experiments conducted in triplicate with similar results. Data in panels C through F are the means ± SD of three independent experiments. Scale bar = 50 *µm*.

statistically significant declines in infectious progeny production and inclusion size (Fig 7A and B).

As the *C. trachomatis* serovar L2 growth rate is relatively fast, with the developmental cycle completed by 48 h (Abdelrahman & Belland, 2005), it can outgrow and/or compensate for certain nutritional stresses (Ouellette et al, 2018). Accordingly, desipramine sensitivity was tested on the slower growing species *C. pneumoniae*, a respiratory pathogen and cause of atherosclerosis that produces infectious progeny by 84 h (Wolf et al, 2000). Laboratory culture of *C. pneumoniae* is fastidious, which can render assessing alterations in drug treatment and infectivity of primary infection difficult. To eliminate the possibility of desipramine pretreatment inhibiting *C. pneumoniae* entry, HeLa cells were first infected with *C. pneumoniae*, and then desipramine or vehicle was added 2–3 h postinfection. Infectious progeny production was measured at 53 and 73 h post-infection. *C. pneumoniae* progeny production was highest late in the

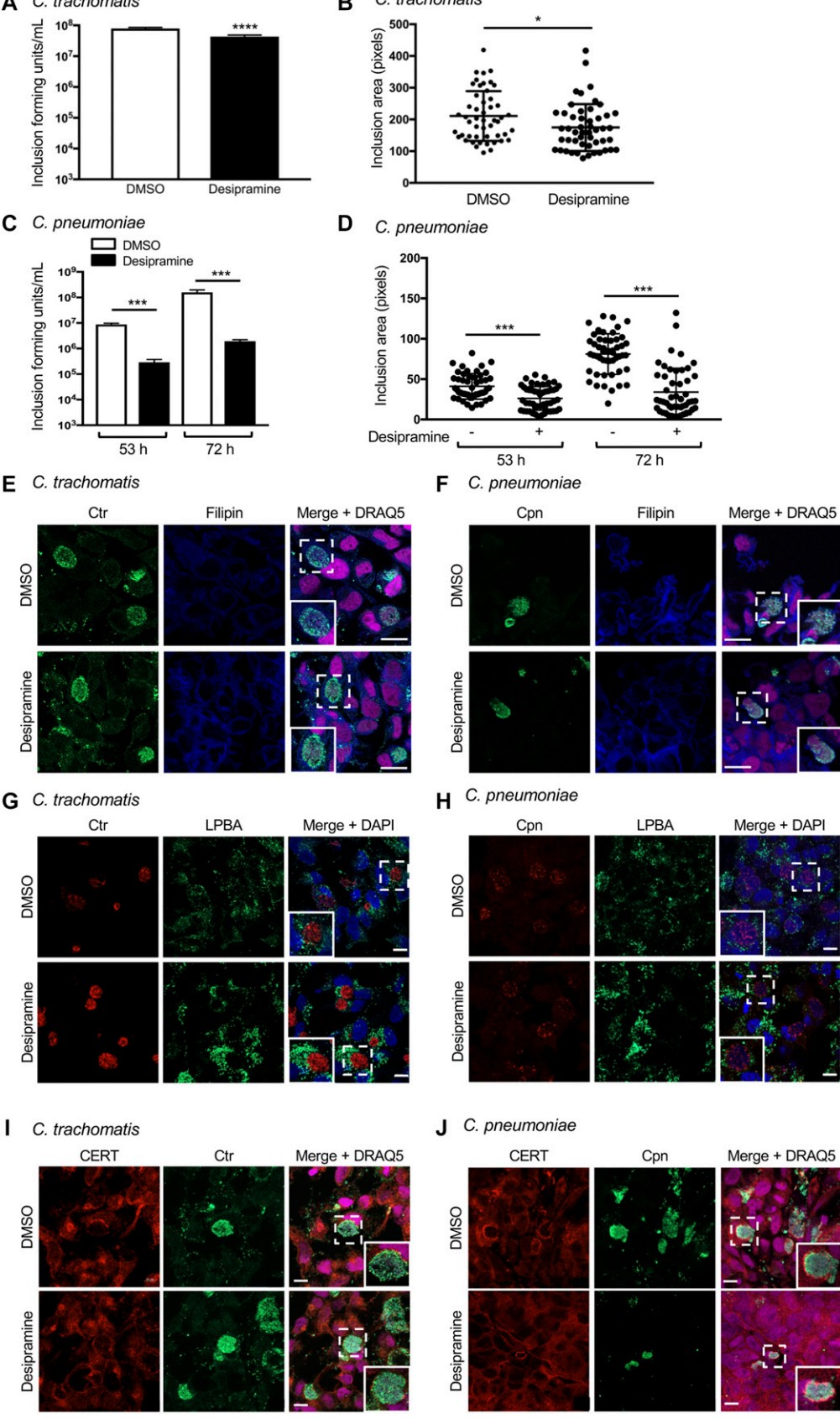

**Figure 7. *C. trachomatis* and *C. pneumoniae* exhibit reduced FIASMA sensitivity.**

**(A–D)** Desipramine or DMSO-treated HeLa cells infected with *C. trachomatis* (Ctr; A, B) or *C. pneumoniae* (Cpn; C, D) were either lysed to recover infectious progeny that were incubated with naïve HeLa cells to determine inclusion-forming units (A, C) or were fixed and assessed by immunofluorescence microscopy to determine inclusion area (B, D). **(E–J)** Desipramine-treated or control cells were infected with Ctr (E, G, I) or Cpn (F, H, J). The cells were either incubated with filipin (E, F) or screened with antibodies specific for LBPA (G, H) or CERT (I, J) together with antisera against *C. trachomatis* (E, G, I) or *C. pneumoniae* (F, H, J). DAPI or DRAQ5 was used to stain DNA. Regions that are demarcated by hatched-line boxes are magnified in the inset panels. Error bars indicate SD. *t* test was used to test for a significant difference among pairs. Statistically significant (*P < 0.05; ***P < 0.001) values are indicated. Data shown are representative of three experiments conducted in triplicate with similar results. Scale bar = 10 μm.

developmental cycle, as illustrated by an almost two-log increase in inclusion-forming units recovered from control cells at 72 h versus 53 h (Fig 7C). In contrast, desipramine significantly reduced infectious progeny production by 1.5 log. Moreover, there was only a 15% increase in progeny production between 53 and 72 h in desipramine-treated cells compared with a 56% increase between these time points in control cells. These data correlate with decreases of approximately 35% in the inclusion area for desipramine-treated versus control cells (Fig 7D). Thus, desipramine effectively inhibits *C. pneumoniae*, but not *C. trachomatis*, infection.

Because the negative impact of desipramine on *C. pneumoniae* was not as severe as for *A. phagocytophilum* and *C. burnetii*, and the drug had no biologically significant effect on *C. trachomatis*, we evaluated the hypothesis that desipramine does not interfere with chlamydial inclusion interactions with the Golgi and MVBs, which, in turn, allows cholesterol delivery to inclusions even when its egress from lysosomes is blocked. Hence, these phenomena were assessed in desipramine-treated and control cells infected with *C. trachomatis* for 24 h or *C. pneumoniae* for 53 h. In accordance with ASM inhibition causing cholesterol to accumulate in lysosomes, desipramine-treated cells displayed vesicle-like structures heavily labeled with filipin (Fig 7E and F). No difference in filipin labeling of chlamydial inclusions was observed. Similar results were noted for lysobisphosphatidic acid (LPBA), an MVB marker that localizes to *C. trachomatis* inclusions (Beatty, 2006, 2008). In desipramine-treated cells, LPBA immunolabeling increased in a similar manner to that observed for cholesterol (Fig 7G and H). LPBA pooled around the inclusions harboring *C. trachomatis*, as previously reported (Beatty, 2006, 2008), and *C. pneumoniae*, observed here for the first time, and did so whether or not the cells were exposed to desipramine. ASM inhibition also had no effect on chlamydial recruitment of VAMP4 and syntaxin 6 (Fig S2), two Golgi-derived soluble N-ethylmalemide–sensitive factor attachment protein receptors that localize to *C. trachomatis* and *C. pneumoniae* inclusions (Moore et al, 2011; Kabeiseman et al, 2013).

Finally, we examined whether localization of ceramide transfer protein (CERT) to inclusions is desipramine-sensitive. CERT, which mediates ER-to-Golgi transfer of ceramide, is recruited to the *C. trachomatis* inclusion membrane, where it contributes to the generation of sphingomyelin, which is important for chlamydial growth (Derre et al, 2011; Elwell et al, 2011). CERT localization to inclusions of both chlamydial species was unhindered by desipramine (Fig 7I and J). This observation marks the first report of CERT being recruited to *C. pneumoniae* inclusions. Altogether, these data demonstrate that the infection cycles of *C. trachomatis* and *C. pneumoniae* are less susceptible to pharmacologic inhibition of ASM than *A. phagocytophilum* or *C. burnetii*, which is due, at least in part, to the abilities of these bacteria to hijack cholesterol from the Golgi and MVBs and to parasitize other lipids such as sphingomyelin.

## Discussion

In this study, we demonstrate that FIASMAs, which are FDA approved, licensed for medical use, exhibit low toxicity, and are suited for prolonged treatment (Kornhuber et al, 2010), reduce host cell infection by four obligate intracellular vacuole-adapted bacteria by blocking LDL cholesterol trafficking from the lysosome. Intracellular pathogens are difficult to target with most conventional antibiotics. Although tetracyclines and fluoroquinolones are usually effective, the occurrence of intracellular bacterial resistance and the drugs' use being contraindicated in certain circumstances warrants development of novel antimicrobial therapies. Finding new therapeutic indications for FDA-approved compounds that disrupt eukaryotic pathways commonly targeted by intracellular bacteria can accelerate drug discovery on this front.

ASM inhibition is most effective against two of the model intracellular bacterial pathogens examined herein, *A. phagocytophilum* and *C. burnetii*, through distinct mechanisms. Desipramine prevents NPC1 trafficking of LDL cholesterol to the ApV, which reversibly blocks inclusion expansion and maturation as well as *A. phagocytophilum* replication and infectious progeny generation. This negative effect was confirmed for two additional FIASMAs and specifically linked to inhibition of ASM, rather than acid ceramidase. Desipramine exerts bacteriostatic action on *A. phagocytophilum* not only when added before, but also when added after bacterial entry, demonstrating the ability to halt active infection. Importantly, this phenomenon is recapitulated in vivo, as the bacterium cannot productively infect ASM-deficient mice or WT mice treated with desipramine within the range approved for human use (Hayasaka et al, 2015).

In addition to intercepting the NPC1 pathway to obtain cholesterol, *A. phagocytophilum* hijacks the Beclin 1-Atg14L autophagy initiation pathway, likely to obtain autophagy-derived amino acids, and TGN exocytic vesicles (Niu et al, 2012; Truchan et al, 2016c). Desipramine inhibits infection, yet Beclin 1, TGN46, and BODIPY ceramide associated with the ApV in treated cells. Thus, although the bacterium's autophagy and TGN parasitism are mutually exclusive from cholesterol acquisition, the latter appears to be most critical for infection cycle progression. Moreover, we previously reported that TGN parasitism is essential for *A. phagocytophilum* transition from noninfectious RCs to infectious DCs (Truchan et al, 2016c). However, the present study reveals that RC-to-DC transition only occurs if NPC1-mediated cholesterol trafficking is not blocked. In the NPC1 pathway, a considerable portion of LDL cholesterol is transported first to the TGN before arrival at the ER, and the TGN is depleted of cholesterol in Niemann–Pick disease (Urano et al, 2008). Thus, desipramine could prevent *A. phagocytophilum*–infectious progeny generation directly by blocking cholesterol-laden NPC1 vesicle delivery to the ApV and indirectly by blocking NPC1 vesicle delivery to the TGN, which, in turn, would culminate in TGN vesicles that are rerouted to the ApV being devoid of cholesterol.

*C. burnetii* is sensitive to cholesterol levels in the CCV, with increased cholesterol further acidifying the vacuole and degrading bacteria (Mulye et al, 2017). Inhibiting ASM with desipramine raises CCV cholesterol levels to kill *C. burnetii* in macrophages and epithelial cells. However, consistent with previous studies using U18666A (Mulye et al, 2017), *C. burnetii* is only sensitive during the first 48 h, which, because the pathogen inhabits a tight-fitting vacuole during the first 24–48 h (Kohler & Roy, 2015), suggests that once the CCV is established and *C. burnetii* is actively growing, either cholesterol no longer has an antimicrobial impact or the bacteria combat the negative effects. Regardless, FIASMAs have

therapeutic potential against Q fever because *C. burnetii* would presumably be susceptible during the early stage of each infection cycle. This potential could extend to other drugs that perturb cholesterol homeostasis, as a recent screen of FDA-approved compounds showed that 57 of 62 tested completely inhibited *C. burnetii* growth, although their mechanisms of action, whether they are bacteriostatic or bactericidal, and whether they are effective during a specific *C. burnetii* infection stage was not determined (Czyz et al, 2014). As demonstrated for both *A. phagocytophilum* and *C. burnetii*, desipramine's antimicrobial efficacy was due exclusively to its ability to promote lysosomal cholesterol sequestration and not a direct effect on either bacterium, suggesting that FIASMAs would likely be invulnerable to the development of pathogen resistance.

Compared with *A. phagocytophilum* and *C. burnetii*, desipramine was less effective against *C. pneumoniae* and even less so against *C. trachomatis* because of chlamydial utilization of not only LDL-, but also non–LDL cholesterol sources and sphingomyelin to fuel intracellular growth. *C. pneumoniae* infections can be persistent, with symptoms often reappearing after a short or conventional course of antibiotics (Burillo & Bouza, 2010). The ability of desipramine to reduce the *C. pneumoniae* load by 1.5 orders of magnitude in vitro suggests that combination drug therapy consisting of an antibiotic plus a FIASMA warrants consideration as an approach to minimize recalcitrant *C. pneumoniae* infections.

Because the susceptibilities of the vacuole-adapted bacteria examined herein to ASM inhibition are linked to sensitivities to lysosomal sequestration of LDL cholesterol, it is reasonable to posit that the antimicrobial efficacy of FIASMAs could be extended to combat other LDL cholesterol–dependent pathogens. *Ehrlichia chaffeensis*, like *A. phagocytophilum*, becomes more infectious when unesterified cholesterol is incorporated into the bacterial cell envelope (Lin & Rikihisa, 2003). As LDL cholesterol only becomes esterified once trafficked to the ER by the NPC1 pathway (Walpole et al, 2018), *E. chaffeensis* presumably intercepts this pathway and would, therefore, be FIASMA sensitive. Similarly, the apicomplexan parasite, *Toxoplasma gondii*, actively intercepts LDL cholesterol, cannot grow in NPC-deficient fibroblasts, and exhibits stunted development in U18666A-treated cells (Coppens et al, 2000). *Brucella abortus* intracellular replication is fairly insensitive to drugs that alter cholesterol homeostasis, but the pathogen requires plasma membrane cholesterol to invade macrophages and cannot infect NPC1 knockout mice (Watarai et al, 2002; Czyz et al, 2014). It is tempting to question whether individuals who are heterozygous for ASM mutations that negatively affect enzymatic activity exhibit resistance to infection by intracellular pathogens that require LDL cholesterol.

Overall, this study identifies FIASMAs as host cell–directed therapeutics for treating infections caused by *A. phagocytophilum*, *C. burnetii*, *C. pneumoniae*, and potentially other pathogens whose infectivity, intracellular growth, and/or survival are strongly influenced by NPC1-trafficked LDL cholesterol. FIASMAs have a high likelihood for preventing resistance development, could be administered as an alternative to antibiotics when necessary or in conjunction with antimicrobial drugs to augment their efficacy, and therefore should be considered for evaluation in clinical settings.

# Materials and Methods

### Cultivation of uninfected and infected cell lines

Uninfected and *A. phagocytophilum* NCH-1 strain–infected human promyelocytic HL-60 cells (CCL-240; American Type Culture Collections [ATCC]) and RF/6A rhesus monkey choroidal endothelial cells (CRL-1780; ATCC) were cultured as previously described (Huang et al, 2010a). HeLa human cervical epithelial cells (CCL-2; ATCC) were maintained as described (Justis et al, 2017). *C. burnetii* Nine Mile Phase II (NMII; clone 4, RSA439) and mCherry-*C. burnetii* NMII were purified from Vero cells (African green monkey kidney epithelial cells; CCL-81; ATCC) or acidified citrate cysteine medium-2 (ACCM-2) and stored as described (Cockrell et al, 2008; Beare et al, 2009). Mouse alveolar macrophages (MH-S; CRL-2019; ATCC) and THP-1 human monocytic cells (TIB-202; ATCC) were maintained as described (Mulye et al, 2018). THP-1 cells were differentiated into macrophage-like cells by overnight treatment with 200 nM phorbol 12-myrisate 13-acetate (MilliporeSigma). *C. trachomatis* serovar L2 (LGV 434) was maintained in HeLa cells at 37°C as described (Rucks et al, 2017). *C. pneumoniae* AR39 was maintained in HeLa cells at 35°C as described (Ouellette et al, 2016). Mammalian cell cultures were low passage and confirmed to be mycoplasma free using the Universal Mycoplasma Detection kit (ATCC) or Mycoplasma PCR Detection kit (MilliporeSigma).

### Antibodies and reagents

Commercial antibodies targeted vimentin (product number ab8069; Abcam), CERT (product number GW22128B; MilliporeSigma), NPC1 (product number ab224268; Abcam), Beclin-1 (product number 207612; Abcam), TGN46 (product number 50595; Abcam), CD63 (product number 556019; BD Biosciences), VAMP4 (product number V4514; MilliporeSigma), syntaxin 6 (product number 610636; BD Biosciences), and LBPA (product number MABT837; MilliporeSigma). Antisera specific for APH0032, APH1235, and P44 were described previously (Huang et al, 2010a; Troese et al, 2011). Antibodies against *C. pneumoniae* were gifts from Ted Hackstadt (National Institute of Allergy and Infectious Diseases [NIAID]; Rocky Mountain Laboratories) or Harlan Caldwell (NIAID; Laboratory of Clinical Immunology and Microbiology). Antibodies against *C. trachomatis* were either a gift from Ted Hackstadt or anti-MOMP (product number C01363; Meridian Life Science). Alexa fluorochrome-conjugated secondary antibodies were obtained from Invitrogen or Jackson ImmunoResearch Laboratories. Filipin (Cayman Chemical) was used to label endogenous cholesterol. LysoTracker Red (Invitrogen), TopFluor (BODIPY) Cholesterol (Avanti Polar Lipids), and BODIPY-TR ceramide (Invitrogen) were used for live cell experiments. DNA stains used were 1,5-bis{[2-(di-methylamino) ethyl]amino}-4, 8-dihydroxyanthracene-9,10-dione (DRAQ5; Thermo Fisher Scientific), DAPI (Thermo Fisher Scientific), and Hoechst 33342 (Thermo Fisher Scientific). Chemical inhibitors used included desipramine (MilliporeSigma), amitriptyline (MilliporeSigma), nortriptyline (MilliporeSigma), and CA-074 Me (MilliporeSigma).

### Isolation of human neutrophils

Human neutrophils were isolated from peripheral blood of healthy donors by centrifugation through an equal volume of Polymorph

Prep (Axis-Shield) at 470 $g$ for 30 min. The resulting neutrophil band was removed via aspiration and mixed with equal volumes of 0.45% (vol/vol) NaCl in PBS and RPMI 1640 (Thermo Fisher Scientific)-0.5 mM EDTA (MilliporeSigma). The cells were centrifuged at 210 $g$ for 10 min and resuspended in Red Blood Cell Lysis Buffer (Thermo Fisher Scientific) for 5 min, washed twice with RPMI 1640-0.5 mM EDTA, and resuspended in RPMI 1640. All investigations using neutrophils obtained from human donor blood were conducted according to the principles expressed in the Helsinki Declaration, and informed consent was obtained from all subjects. The protocol (HM11407) for obtaining donor blood for the purpose of isolating neutrophils has been reviewed and approved by the Virginia Commonwealth University Institutional Review Board with respect to scientific content and compliance with applicable research and human subject regulations.

### Infection assays

For *A. phagocytophilum* infections, HL-60 cells, RF/6A cells, or human neutrophils were treated for 1 h with 1–10 $\mu$M desipramine, 10 $\mu$M amitriptyline, 10 $\mu$M nortriptyline, or DMSO before incubation with *A. phagocytophilum* DC organisms as described (Troese & Carlyon, 2009; Truchan et al, 2016a, 2016c) in the continued presence of FIAMSA unless otherwise noted. In some instances, HL-60 cells were infected before desipramine treatment or *A. phagocytophilum* DC organisms were treated with desipramine before incubation with host cells, whereas in others desipramine was either added to or removed from *A. phagocytophilum*−infected HL-60 cells at 20 h. To confirm that desipramine's inhibitory effect on *A. phagocytophilum* was specific to its action on ASM and not acid ceramidase, HL-60 cells were pretreated with 5 $\mu$M CA-074 Me before desipramine treatment and *A. phagocytophilum* infection.

For *C. burnetii* infections, MH-S or HeLa cells were treated for 1 h with desipramine or DMSO and infected as previously described with mCherry-expressing small cell variants for 1 h (Mulye et al, 2017; Clemente et al, 2018). Infection conditions were optimized for both cell type and vessel for less than one internalized bacterium per cell. THP-1 macrophage-like cells were infected with mCherry-expressing *C. burnetii* at a multiplicity of infection (MOI) of 10 in the presence of 10 $\mu$M desipramine or DMSO.

For *C. trachomatis* infections, HeLa cells were treated for 1 h with 10 $\mu$M desipramine or DMSO, then infected at an MOI of 0.5 by adding inoculum to tissue culture wells and centrifuging the tissue culture plates at 400 $g$ for 15 min at room temperature, followed by incubation at 37°C, 5% $CO_2$, for the indicated time points. For *C. pneumoniae* infections, HeLa cells were inoculated at an MOI of 2 as described for *C. trachomatis* infections with the exception that infected monolayers were incubated at 35°C, 5% $CO_2$. At 2–3 h postinfection, the medium was changed and 10 $\mu$M desipramine or DMSO was added. To assess chlamydial-infectious progeny production, infected monolayers were scraped and centrifuged at 17,000 $g$ for 30 min at 4°C. The pellets were resuspended in sucrose phosphate buffer and vortexed in the presence of glass beads. Resulting lysates were serially diluted and plated onto fresh HeLa cells in the absence of desipramine. Monolayers were centrifuged at 400 $g$ for 15 min at room temperature. *C. trachomatis*−infected cells were placed at 37°C, 5% $CO_2$, and incubated for an additional

24 h. *C. pneumoniae*−infected cells were placed at 35°C, 5% $CO_2$, and incubated for an additional 64 h. For *A. phagocytophilum*, *C. burnetii*, and chlamydial infection experiments in which desipramine treatment extended beyond 24 h, fresh medium containing desipramine or DMSO was added to the cultures every 24 h.

### Immunofluorescence microscopy

For *A. phagocytophilum* immunofluorescence assays, infected RF/6A cells on 12-mm glass coverslips (Electron Microscopy Sciences) were fixed in 4% (vol/vol) PFA (Electron Microscopy Sciences) in PBS for 20 min followed by permeabilization with 0.5% (vol/vol) Triton X-100 in PBS for 15 min. Immunofluorescence labeling was performed as previously described (Huang et al, 2010a) followed by DAPI staining of DNA and mounting with Prolong Gold antifade reagent (Thermo Fisher Scientific). Images were obtained at room temperature using a Zeiss LSM 700 laser-scanning confocal microscope (Zeiss) and a 63× oil-immersion objective with a 1.4 numerical aperture. Images were acquired using Zeiss Efficient Navigation Imaging Suite 2.3 Blue Edition. For *C. burnetii* immunofluorescence assays, infected HeLa cells on 12-mm glass coverslips were fixed with 2.5% (vol/vol) PFA in PBS for 15 min, followed by permeabilization and blocking in PBS containing 0.1% (vol/vol) saponin and 1% BSA. Immunofluorescence staining with CD63 antibody was performed as previously described (Justis et al, 2017) and mounted with Prolong Gold antifade reagent. Images were obtained at room temperature using a Nikon TiE inverted microscope with 60× oil immersion objective having a 1.4 numerical aperture and an ORCA-Flash 4.0 LT + sCMOS camera (Hammamatsu). Chlamydial infected cells were imaged at room temperature using a Zeiss LSM 810 laser-scanning confocal microscope with a 63× oil immersion objective having a 1.4 numerical aperture. Images were acquired using Zeiss Efficient Navigation Imaging Suite 2.3 Blue Edition. Inclusion-forming units were calculated using an Olympus CKX53 microscope with a 40× objective. Inclusion-forming units per mL were determined by calculating the average number of inclusions per field of view multiplied by the dilution factor and the number of fields of view per well, divided by the volume of the original inoculum. The number of fields of view per well was determined by dividing the surface area of the tissue culture well by the view area of the field.

### Fluorescent analogue labeling

For filipin labeling, infected cells were fixed in 2.5% (vol/vol) PFA in PBS on ice for 15 min and incubated with filipin in 1% (vol/vol) BSA in PBS for 1 h as previously described (Mulye et al, 2017). The cells were mounted with Prolong Gold antifade reagent lacking DAPI and imaged via confocal microscopy. For BODIPY cholesterol labeling, live *A. phagocytophilum*−infected RF/6A cells were incubated with 4 mg ml$^{-1}$ BODIPY cholesterol in cholesterol-free medium for a minimum of 18 h to allow the cholesterol to accumulate in lysosomes and washed with Hepes (Thermo Fisher Scientific) twice. The cells were then incubated with BODIPY ceramide in Hepes per the manufacturer's instructions and stained with Hoechst 33342 (Invitrogen) to label host cell nuclei and bacteria. In some instances, live cells were incubated with LysoTracker Red in media for

3 h to identify acidic compartments. The live cells were then imaged at room temperature with a Leica TCS SP8 microscope (Leica) affixed with an Andor iXon Life 888 EMCCD camera (Oxford Instruments) and a 63× water-immersion objective with a 1.2 numeric aperture. Images were acquired using Leica Application Suite X software. The brightness of post-acquisition images of RF/6A cells labeled with LysoTracker Red, BODIPY cholesterol, and Hoechst 33342 was increased using PowerPoint version 16.16.3 (Microsoft).

### qPCR and qRT-PCR

To analyze the *A. phagocytophilum* load after inhibitor treatment, DNA was isolated from the infected cells with the DNeasy Blood and Tissue kit (QIAGEN). Bacterial load in infected HL-60 and RF/6A cells was determined using primers specific for *A. phagocytophilum* 16S rDNA and *β*-actin (Oki et al, 2016), the latter of which target conserved sequences among human, primate, and murine *β*-actin, SsoFast EvaGreen Supermix (Bio-Rad), and 100 ng template DNA. Thermal cycling conditions used were 98°C for 2 min, followed by 40 cycles of 98°C for 5 s and 55°C for 10 s. Relative 16S rDNA was normalized to *β*-actin using the $2^{-\Delta\Delta CT}$ (Livak) method (Livak & Schmittgen, 2001). Total RNA isolated from *A. phagocytophilum*–infected HL-60 cells using the RNeasy RNA Isolation kit (QIAGEN) was treated with DNase I (Invitrogen) and used as template for cDNA synthesis with the iScript Reverse Transcription Supermix (Bio-Rad). 1 *µ*l of a 1:10 dilution of the cDNA was used as a template for qRT-PCR as described (Kahlon et al, 2013) using primers specific for *β*-actin, *A. phagocytophilum* 16S rDNA, and *aph1235* (Troese et al, 2011). Thermal cycling conditions used were 98°C for 30 s, followed by 40 cycles of 98°C for 5 s and 60°C for 5 s. Relative *aph1235* transcript levels were normalized to 16S rRNA levels using the $2^{-\Delta\Delta CT}$ method.

### *C. burnetii* CFU assay

Viable *C. burnetii* were assayed using a CFU assay as previously described (Clemente et al, 2018). Briefly, bacteria were recovered from infected cells using water lysis as described (Mulye et al, 2017). The released bacteria were serially diluted in ACCM-2 and spotted in duplicate onto 0.25% ACCM-2 with tryptophan agarose plates (Vallejo Esquerra et al, 2017). The plates were incubated for 9–12 d at 37°C, 2.5% $O_2$, and 5% $CO_2$; Colony numbers were determined to measure viable bacteria. Each experiment was performed in biological duplicate.

### Mouse studies

ASM$^{-/-}$ mice were a gift from Pin-Lan Li (Virginia Commonwealth University, Richmond, VA). 6–8-wk-old ASM$^{-/-}$ or C57Bl/6 male mice were injected intraperitoneally with $10^8$ *A. phagocytophilum* DC organisms as described (Naimi et al, 2018). Male mice were exclusively used because they are more susceptible to *A. phagocytophilum* infection than female mice (Naimi et al, 2018). For desipramine treatment studies, 6–8-wk-old C57Bl/6 male mice were intraperitoneally injected twice per day with either 10 mg·kg$^{-1}$ desipramine or PBS on days 7–12 post *A. phagocytophilum* infection as described (Teichgraber et al, 2008). DNA was isolated from blood

collected via the tail vein on days 4, 8, 12, 16, 21, and 28 postinfection using the DNeasy Blood and Tissue kit (QIAGEN). The peripheral blood *A. phagocytophilum* load was determined by qPCR as described above except that 50 ng of DNA was used as template. Thermal cycling conditions used were 98°C for 2 min, followed by 40 cycles of 98°C for 5 s and 60°C for 30 s. Mice were euthanized on day 28. All animal research was conducted in compliance with the Health and Human Services Guide for the Care and Use of Laboratory Animals and performed under the approval of the Institutional Animal Care and Use Committee at Virginia Commonwealth University (Protocol AM10220).

### Statistical analyses

Statistical analyses were performed using the Prism 5.0 software package (GraphPad). One-way ANOVA with Tukey's post hoc test was used to test for a significant difference among groups. *t* test was used to test for a significant difference among pairs. Statistical significance was set at *P* values of < 0.05. Vacuole size measurements were done with ImageJ (W. S. Rasband, National Institutes of Health) (Schindelin et al, 2012).

## Supplementary Information

## Acknowledgements

We thank Joao F Pedra (University of Maryland, Baltimore, MD), Robert A Heinzen (NIAID, Hamilton, MT), and Ted Hackstadt (NIAID, Hamilton, MT) for critical review of this manuscript. We thank Waheeda A Naimi for technical assistance. Confocal micrographs of *A. phagocytophilum*–infected cells were obtained at the VCU Microscopy Facility, which is supported, in part, with funding from National Institutes of Health - National Institute of Neurological Disorders and Stroke (NIH-NINDS) Center core grant 5P30NS047463 and NIH-NCI Cancer Center Support Grant (P30 CA016059). Chlamydial images were acquired with the assistance of Janice A Taylor and James R Talaska in the University of Nebraska Medical Center Advanced Microscopy Core Facility, which is funded by the Fred and Pamela Buffet Cancer Center Support Grant (P30CA036727) and an Institutional Development Award from National Institute of General Medical Sciences (NIGMS) of the NIH (P30GM106397). JA Carlyon is supported by National Institutes of Health - National Institute of Allergy and Infectious Diseases (NIH-NIAID) grants 1R01 AI139072 and 2R01 AI072683. CE Chalfant is supported by NIH-NIAID grant 1R01 AI139072. DH Conrad is supported by NIH-NIAID grant 5R01 AI018697. DE Voth is supported by NIH-NIAID grant 1R21 AI127931 and NIH/NIGMS grant 1P20 GM103625. EA Rucks is supported by 1R01 AI114670 and University of Nebraska Medical Center (UNMC) start-up funds. SD Gilk is supported by NIH-NIAID grants R21AI121786 and R01AI139176.

### Author Contributions

CL Cockburn: conceptualization, data curation, formal analysis, supervision, investigation, methodology, and writing—original draft, review, and editing.
RS Green: data curation, formal analysis, investigation, and writing—review and editing.

SR Damle: data curation, investigation, and writing—review and editing.

RK Martin: data curation, investigation, and writing—review and editing.

NN Ghahrai: data curation, investigation, and writing—review and editing.

PM Colonne: data curation, investigation, and writing—review and editing.

MS Fullerton: data curation, investigation, and writing—review and editing.

DH Conrad: formal analysis, supervision, funding acquisition, validation, and writing—review and editing.

CE Chalfant: conceptualization, formal analysis, funding acquisition, and writing—review and editing.

DE Voth: conceptualization, resources, formal analysis, supervision, funding acquisition, methodology, and writing—original draft, review, and editing.

EA Rucks: conceptualization, resources, data curation, formal analysis, funding acquisition, investigation, methodology, and writing—original draft, review, and editing.

SD Gilk: conceptualization, resources, data curation, formal analysis, funding acquisition, investigation, methodology, and writing—original draft, review, and editing.

J Carlyon: conceptualization, resources, data curation, formal analysis, supervision, funding acquisition, validation, methodology, project administration, and writing—original draft, review, and editing.

## Conflict of Interest Statement

The authors declare that they have no conflict of interest.

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
