## [Reviewer comments · Life Science Alliance]

Life Science Alliance

Functional inhibition of acid sphingomyelinase disrupts infection by intracellular bacteria

Chelsea Cockburn, Ryan Green, Sheela Damle, Rebecca Martin, Naomi Ghahrai, Punsiri Colonne, Marissa Fullerton, Daniel Conrad, Charles Chalfant, Daniel Voth, Elizabeth Rucks, Stacey Gilk, and Jason Carlyon

DOI: <https://doi.org/10.26508/lsa.201800292>

Corresponding author(s): Jason Carlyon, Virginia Commonwealth University

Review Timeline:

Submission Date:	2018-12-30
Editorial Decision:	2019-02-07
Revision Received:	2019-03-05
Editorial Decision:	2019-03-07
Revision Received:	2019-03-12
Accepted:	2019-03-13

Scientific Editor: Andrea Leibfried

Transaction Report:

February 7, 2019

Re: Life Science Alliance manuscript #LSA-2018-00292-T

Dr. Jason Carlyon
Virginia Commonwealth University
Box 980678
Richmond, Virginia 23298-0678

Dear Dr. Carlyon,

Thank you for submitting your manuscript entitled "Functional inhibition of acid sphingomyelinase disrupts infection by intracellular bacterial pathogens" to Life Science Alliance. The manuscript was assessed by expert reviewers, whose comments are appended to this letter.

As you will see, the reviewers appreciate your analysis and provide constructive input on how to further strengthen the data presentation and discussion / extension to epidemiological data in a minor revision. I would thus like to invite you to provide a revised version of your manuscript, addressing the points made by the reviewers.

Thank you for this interesting contribution to Life Science Alliance. We are looking forward to receiving your revised manuscript.

Sincerely,

Andrea Leibfried, PhD
Executive Editor

Life Science Alliance
Meyerhofstr. 1
69117 Heidelberg, Germany
t +49 6221 8891 502
e a.leibfried@life-science-alliance.org
www.life-science-alliance.org

- A letter addressing the reviewers' comments point by point.
- An editable version of the final text (.DOC or .DOCX) is needed for copyediting (no PDFs).
- High-resolution figure, supplementary figure and video files uploaded as individual files: See our detailed guidelines for preparing your production-ready images, <http://life-science-alliance.org/authorguide>
- Summary blurb (enter in submission system): A short text summarizing in a single sentence the study (max. 200 characters including spaces). This text is used in conjunction with the titles of papers, hence should be informative and complementary to the title and running title. It should describe the context and significance of the findings for a general readership; it should be written in the present tense and refer to the work in the third person. Author names should not be mentioned.

B. MANUSCRIPT ORGANIZATION AND FORMATTING:

Full guidelines are available on our Instructions for Authors page, <http://life-science-alliance.org/authorguide>

Reviewer #2 (Comments to the Authors (Required)):

The manuscript entitled "Functional inhibition of acid sphingomyelinase disrupts infection by intracellular bacterial pathogens" by Chelsea L. Cockburn et al. describes that inhibition of the acid sphingomyelinase by functional inhibitors of the enzyme regulates cholesterol egress from the lysosome and thereby the infection cycles of vacuole-adapted bacteria. The authors apply these concepts to cellular infection with *Anaplasma phagocytophilum*, *Coxiella burnetii* and *Chlamydiae* and demonstrate a marked inhibition of the infection cycles and/or an indirect bactericidal effect of

functional acid sphingomyelinase inhibitors on these pathogens. They extend these studies to in vivo infections and demonstrate that *A. phagocytophilum* fails to productively infect in acid sphingomyelinase deficient mice or after treatment of wildtype mice with acid sphingomyelinase inhibitors.

The authors provide an impressive body of work showing inhibition of growth of (some) intracellular bacteria. They address and identify cholesterol flux as critical target for the effects of acid sphingomyelinase inhibitors on bacterial growth.

The authors also provide impressive in vivo data using acid sphingomyelinase-deficient mice.

The data are of high quality and the experiments are well performed.

The study identifies a novel function of the acid sphingomyelinase and provides convincing evidence for the ability to treat infections with *Anaplasma phagocytophilum*, *Coxiella burnetii* and *Chlamydiae* with functional inhibitors of the acid sphingomyelinase. This is of broad interest.

The study is very complete and I only have a few suggestions:

1. It would be interesting to determine the residual activity of the acid sphingomyelinase after treatment with desipramine. These drugs usually do not induce complete inhibition of the enzyme and it would be important to determine the degree of acid sphingomyelinase inhibition that is required to inhibit the infections.
2. In line with the degree of acid sphingomyelinase inhibition that is sufficient to prevent bacterial growth, the authors may want to discuss whether individuals who are heterozygous for acid sphingomyelinase mutations and, thus, healthy, might be protected from infections with certain pathogens and whether this provides a survival advantage.
3. Desipramine and also many similar drugs such as amitriptyline, imipramine, fluoxetine, etc. are widely used as antidepressants. Are any data available that patients that are taking these drugs are protected from the infection with intracellular pathogens?

Reviewer #3 (Comments to the Authors (Required)):

The authors present the relevant topic of intracellular bacterial pathogens and provide mechanistic evidence how functional inhibitors of acid sphingomyelinase (ASM) could serve in conjunction or as alternative to antibiotics to combat these infections. Evidence is collected in different relevant cell types for four vacuole-adapted bacteria as models - *Anaplasma phagocytophilum*, *Coxiella burnetii*, *Chlamydia trachomatis*, and *Chlamydia pneumoniae* - with varying degree of dependence on cholesterol trafficking and also extended to the application in mice. The effects of ASM inhibition by treatment with desipramine (amitriptyline and nortriptyline as confirmation) are shown particularly for *A. phagocytophilum* with respect to the generation of bacterial load, infectious progeny, vacuole maturation and expansion, and the stage of the infection cycle and are compared to those in the other model species.

The authors introduce the topic with recent literature and provide sufficient background.

Experiments contain mostly clear method descriptions including controls and appear to present solid results; the data are convincing, documented in detail with additional test and well visualized in informative figures. The results section frequently contains background explanations that could be transferred to the introduction but due to the utilization of four different models are helpful to understand the experiments and data at their place. The discussion is well prepared.

There are only a number of minor comments and suggestions (see below). Some additional questions to consider: Is there any data on a lower susceptibility of Niemann-Pick Disease patients to infections with the mentioned group of bacteria? How about other lysosomal storage diseases, is infectivity altered, are there epidemiological data? Or is there any information on susceptibility of patients taking medication that acts as FIASMAs such as many antidepressants - are these people protected to some degree?

Minor comments:

Introduction:

Lines 66-67: "Niemann-Pick disease severity correlates with decreased ASM activity (Kornhuber et al., 2010)." Why has this reference on FIASMAs instead of one on NPD (original work) been chosen here?

Lines 67-71: "Conversely, 68 ASM activation has also been linked to development of multiple human diseases 69 [reviewed in (Kornhuber et al., 2010; Schuchman, 2010)],... have beneficial consequences (Kornhuber et al., 2010)" Why has the reference Kornhuber et al. 2010 on FIASMAs been chosen here instead of, for example, the review Kornhuber J, Rhein C, Muller CP, Muhle C (2015) Secretory sphingomyelinase in health and disease. *Biological chemistry* 396 (6-7):707-736. doi:10.1515/hsz-2015-0109

Lines 80-83: "FIASMA could represent novel, non-antibiotic means for treating diverse infectious diseases. Yet, their potential in this capacity, and the importance of ASM in intracellular bacterial infection has gone largely unexplored."

Why is some existent work on this topic not cited here? For example work of the Gulbins / Grassmé groups on *Staphylococcus aureus* or *Mycobacterium tuberculosis* (even though the mechanism would be different):

Peng et al. (2015): Acid sphingomyelinase inhibition protects mice from lung edema and lethal *Staphylococcus aureus* sepsis

Wu et al. (2018): The function of sphingomyelinases in mycobacterial infections

Results

For figures in general, it could be helpful to include a label of the cells applied (HL-60 vs. RF/6A) or a label of the bacterial species (e.g. Figures 7 A+B vs. C+D) to see this information quickly without searching through the legend.

Lines 100-113: This part does not report results but is rather an introductory paragraph on A. phagocytophilum and should thus be moved to the introduction section.

Lines 114 + 119 state that "Promyelocytic HL-60 and RF/6A endothelial cells" were used in this part but the figure 1 legend only names "HL-60 cells ... or human peripheral blood neutrophils". It appears that the RF/6A cell data are shown in S1 but also some important HL-60 data. There is continuous switching between these two figures. It would be easier to follow if all relevant data (figure 1 and S1) were combined into one larger figure with a common legend. The amount of details of the two legends is not consistent (also for later figures).

Figure 1:

For faster grasping the information, it could be worthwhile to leave the first bar (DMSO) in Figure 1B empty (instead of full black color) to have this comparable to all other DMSO conditions in Figure 1.

Figure 2: The legend states, cells were fixed at 20, 24, 28 and 32 h in A to C, but figure A shows no 32h data - what is the reason?

Labeling of figures 2D-E-F could be improved and shortened by showing the times below the identical time points (20h etc.) and for example +/- for desipramine maintenance /removal of full and empty circle symbols with a legend for desipramine maintained or removed €, similarly for D (empty vs. full symbols) and E. It would be better to have D-E-F aligned in a column (for example swap figures C and D). The legend describes (lines 959-960) the maintenance (D) and removal (E) of desipramine but not (F) with later addition of desipramine.

Lines 178-180: "Vimentin associated with all ApVs observed under both conditions (Figure 2A), indicating that this early ApV biogenesis event is not dependent on ASM activity." How do the authors prove that vimentin (despite its early recruitment and irreversible association with ApV lines described in lines 169-170) is associated with all (?) ApVs?

Line 284 "Beginning on day 3, the first day on which there was any detectable increase in fluorescence signal..." - In figure 6, to which conditions are the time points compared (indicated by asterisks)? It is not obvious that there is already a clear increase at day 3 labelled "****" compared to day 0. Explain abbreviation Cb (*C. burnetii*) in the legend.

Line 376 explain abbreviation NSF

Methods:

Lines 501f: To allow replication of presented results, used commercial antibodies should be listed with their product numbers not just the company, since many companies offer different antibodies for the same target.

Lines 538f: For how long were the cells left with the FIASMA: for the entire duration of the infection or just the 1h prior to infection? The exchange of media (line 567f) suggests the latter but it's not clear from the sentence.

Line 550f: "Infection conditions were optimized for both cell type and vessel for less than one internalized bacterium per cell." Is this true only for the infections with *A. phagocytophilum* described above or only for the infections described below the sentence or for both? If for all, consider adding "all" and moving the sentence.

Line 576: "0.5% Triton X-100" in PBS as given for PFA?

Line 584: "0.1% saponin in 1% BSA" in PBS?

Line 601: "2.5% PFA" also in PBS as before?

Line 602 "1% BSA" also in PBS?

Please always consistently state the exact solution or provide a general sentence (all in PBS).

Line 607 and later: "Hepes (Thermo Fisher Scientific)", please use HEPES buffer

Lines 622f: "Bacterial load was determined using primers specific for *A. phagocytophilum* 16S rDNA and host cell beta-actin using SsoFast EvaGreen Supermix (BioRad, Hercules, CA) as previously described (Ojogun et al., 2012)." Please provide exact primers used in this study. The reference Ojogun et al. 2012 lists primers for RT-qPCR including Ap 16S-527F and Ap 16S-753R that could be meant but none for (species specific to the used cells) beta-actin.

"Relative 16S rDNA was normalized to mouse beta-actin using the 2-DeltaDeltaCT 626 (Livak) method" - why mouse? Cell lines were not (all) from mice. Does this only refer to mouse blood samples?

Was RNA quality checked?

It would be helpful to provide approximate amounts of template/dilutions used for these reactions

(qPCR, cDNA synthesis, RT-qPCR). In general, MIQE guidelines should be followed where possible.

Lines 657f: "qPCR as described above. Thermal cycling conditions used were 98{degree sign}C for 2 min, followed by 40 cycles of 98{degree sign}C for 5 s and 60{degree sign}C for 30 s" Why was the thermal cycling protocol different from the one described above? Which beta-actin primers have been used for the mouse studies?

Language

Please check spelling and correct notation particularly of chemicals throughout the text, for example:

Line 517f: Spelling mistakes:

Hoeschst 33342 should be Hoechst

amitryptiline (MilliporeSigma) should be amitriptyline

nortryptiline (MilliporeSigma) should be nortriptyline

Ca074-Me (MilliporeSigma) should be CA-074 Me

Bodipy (e.g. figures 4D, E) should be consistently BODIPY

March 5, 2019

School of Medicine

Jason A. Carlyon, Ph.D., Professor
Department of Microbiology & Immunology
Box 980678
Virginia Commonwealth University
Richmond, Virginia 23298-0678
804-628-3382; Fax: 804-828-9946
E-mail: jason.carlyon@vcuhealth.org

RE: Response to reviewer comments for manuscript LSA-2018-00292-TR

Dear Reviewers:

We hereby resubmit our manuscript, "Functional inhibition of acid sphingomyelinase disrupts infection by intracellular bacterial pathogens" (LSA-2018-00292-TR) for your consideration for publication in *Life Science Alliance*. Thank you for the opportunity to do so. We are encouraged by your enthusiasm for the studies and appreciate your efforts in identifying areas for improvement. Of the 35 points raised, 33 were editorial, each of which we addressed. We performed the experiment requested by Reviewer 3 and include the results as new supplementary data (Figure S1A). Reviewer 2 stated that it would be interesting to perform a particular experiment, but this experiment was not pursued due to numerous factors that we respectfully submit would have confounded interpretation of the results. Overall, we believe that the revised manuscript has been considerably strengthened and hope that you will now view it as acceptable for publication. Below is our point-by-point response to your critiques. Thank you for your consideration.

Sincerely,

Jason A. Carlyon, Ph.D.
Professor

Reviewer #2:

The manuscript entitled "Functional inhibition of acid sphingomyelinase disrupts infection by intracellular bacterial pathogens" by Chelsea L. Cockburn et al. describes that inhibition of the acid sphingomyelinase by functional inhibitors of the enzyme regulates cholesterol egress from the lysosome and thereby the infection cycles of vacuole-adapted bacteria. The authors apply these concepts to cellular infection with *Anaplasma phagocytophilum*, *Coxiella burnetii* and *Chlamydiae* and demonstrate a marked inhibition of the infection cycles and/or an indirect bactericidal effect of functional acid sphingomyelinase inhibitors on these pathogens. They extend these studies to in vivo infections and demonstrate that *A. phagocytophilum* fails to productively infect in acid sphingomyelinase deficient mice or after treatment of wildtype mice with acid sphingomyelinase inhibitors.

The authors provide an impressive body of work showing inhibition of growth of (some) intracellular bacteria. They address and identify cholesterol flux as critical target for the effects of acid sphingomyelinase inhibitors on bacterial growth. The authors also provide impressive in vivo data using acid sphingomyelinase-deficient mice.

The data are of high quality and the experiments are well performed. The study identifies a novel function of the acid sphingomyelinase and provides convincing evidence for the ability to treat infections with *Anaplasma phagocytophilum*, *Coxiella burnetii* and *Chlamydiae* with functional inhibitors of the acid sphingomyelinase. This is of broad interest. The study is very complete and I only have a few suggestions:

1. It would be interesting to determine the residual activity of the acid sphingomyelinase after treatment with desipramine. These drugs usually do not induce complete inhibition of the enzyme and it would be important to determine the degree of acid sphingomyelinase inhibition that is required to inhibit the infections.

RESPONSE: We strive to be as accommodating as possible, but respectfully submit that this experiment would be confounded by the following issues. In addition to ASM that remains in the lysosome, a portion of the ASM pool is translocated to the plasma membrane, and there is also secretory ASM. FIASMAs accumulate in lysosomes to specifically inhibit ASM in that organelle, which blocks cholesterol egress to the detriment of *A. phagocytophilum*, *C. burnetii*, and *C. pneumoniae*. Secretory and plasma membrane-localized ASM are not inhibited by FIASMAs. Because commercially available ASM enzymatic assays (e.g. Abcam product number ab190554; ThermoFisher product number A12220) do not discriminate between lysosomal ASM activity from that of secretory ASM or plasma membrane-localized ASM, it would be impossible to correlate ASM activity with effects on intracellular bacteria. Additionally, it is well-established that many intracellular microbes (e.g. *Brucella*, *Neisseria*, ebola virus, sindbis virus, rhinoviruses) utilize ASM as a receptor and/or exploit its activity to enter host cells. While it is not yet known if any of the model pathogens utilized in the current study do so, there is at least circumstantial published evidence that *A. phagocytophilum* does enter host cells at lipid rafts (plasma membrane sites where ASM is known to be enriched). Thus, it is reasonable to posit that the organisms studied herein might exploit or modulate plasma membrane activity, which could also confound the assay. Please note that, while due to these challenges, we did not pursue this suggested experiment, we did perform the only other suggested experiment, which was proposed by Reviewer 3.

2. In line with the degree of acid sphingomyelinase inhibition that is sufficient to prevent bacterial growth, the authors may want to discuss whether individuals who are heterozygous for acid sphingomyelinase mutations and, thus, healthy, might be protected from infections with certain pathogens and whether this provides a survival advantage.

RESPONSE: We have added such a statement to the Discussion. Please see lines 469-471.

3. Desipramine and also many similar drugs such as amitriptyline, imipramine, fluoxetine, etc. are widely used as antidepressants. Are any data available that patients that are taking these drugs are protected from the infection with intracellular pathogens?

RESPONSE: This is a great and insightful question. To our knowledge, the answer is no. But, we are very interested in pursuing. In fact, Dr. Cockburn (first author of this manuscript), who is returning to medical school to continue her clinical rotations in Spring 2019, has an internal grant under review by a University committee to pursue this project as part of her medical school studies. If funded, I would remain as her advisor for this project.

Reviewer #3:

The authors present the relevant topic of intracellular bacterial pathogens and provide mechanistic evidence how functional inhibitors of acid sphingomyelinase (ASM) could serve in conjunction or as alternative to antibiotics to combat these infections. Evidence is collected in different relevant cell types for four vacuole-adapted bacteria as models - *Anaplasma phagocytophilum*, *Coxiella burnetii*, *Chlamydia trachomatis*, and *Chlamydia pneumoniae* - with varying degree of dependence on cholesterol trafficking and also extended to the application in mice. The effects of ASM inhibition by treatment with desipramine (amitriptyline and nortriptyline as confirmation) are shown particularly for *A. phagocytophilum* with respect to the generation of

bacterial load, infectious progeny, vacuole maturation and expansion, and the stage of the infection cycle and are compared to those in the other model species.

The authors introduce the topic with recent literature and provide sufficient background. Experiments contain mostly clear method descriptions including controls and appear to present solid results; the data are convincing, documented in detail with additional test and well visualized in informative figures. The results section frequently contains background explanations that could be transferred to the introduction but due to the utilization of four different models are helpful to understand the experiments and data at their place. The discussion is well prepared.

1. There are only a number of minor comments and suggestions (see below). Some additional questions to consider: Is there any data on a lower susceptibility of Niemann-Pick Disease patients to infections with the mentioned group of bacteria? How about other lysosomal storage diseases, is infectivity altered, are there epidemiological data? Or is there any information on susceptibility of patients taking medication that acts as FIASMAs such as many antidepressants - are these people protected to some degree?

RESPONSE: The minor comments have been addressed point-by-point below. In regards to Niemann-Pick Disease patients or individuals taking FIASMAs being resistant to certain intracellular bacterial infections, to our knowledge such literature is lacking. But, this is an exciting line for future investigation. Per our response to Reviewer 2's third query, Dr. Cockburn (first author of this manuscript), who is returning to medical school to continue her clinical rotations in Spring 2019, has an internal grant under review by a University committee to pursue this project as part of her medical school studies. If funded, I would remain as her advisor for this project.

Minor comments:

Introduction:

2. Lines 66-67: "Niemann-Pick disease severity correlates with decreased ASM activity (Kornhuber et al., 2010)." Why has this reference on FIASMAs instead of one on NPD (original work) been chosen here?

RESPONSE: We have replaced the Kornhuber et al 2010 reference with the original reference that Kornhuber cited for this statement (Schumann and Miranda 1997; please see line 66). We have also cited the first report of Niemann-Pick disease (Brady 1966) at the first sentence of the paragraph (please see line 61).

3. Lines 67-71: "Conversely, 68 ASM activation has also been linked to development of multiple human diseases 69 [reviewed in (Kornhuber et al., 2010; Schuchman, 2010)],... have beneficial consequences (Kornhuber et al., 2010)" Why has the reference Kornhuber et al. 2010 on FIASMAs been chosen here instead of, for example, the review Kornhuber J, Rhein C, Muller CP, Muhle C (2015) Secretory sphingomyelinase in health and disease. *Biological chemistry* 396 (6-7):707-736. doi:10.1515/hsz-2015-0109

RESPONSE: We agree that the more recent Kornhuber review should have been cited here. The replacement has been made. Please see line 68.

4. Lines 80-83: "FIASMAs could represent novel, non-antibiotic means for treating diverse infectious diseases. Yet, their potential in this capacity, and the importance of ASM in intracellular bacterial infection has gone largely unexplored."

Why is some existent work on this topic not cited here? For example work of the Gulbins / Grassmé groups on *Staphylococcus aureus* or *Mycobacterium tuberculosis* (even though the mechanism would be different): Peng et al. (2015): Acid sphingomyelinase inhibition protects mice from lung edema and lethal *Staphylococcus aureus* sepsis

Wu et al. (2018): The function of sphingomyelinases in mycobacterial infections

RESPONSE: We expand the introduction to include mention of the studies on ASM/FIASMA treatment in the context of *Staphylococcus aureus* and mycobacterial infections. We have referenced the two recommended

papers as well as two original studies pertaining to mycobacteria referenced by Wu et al 2018. Please see lines 77-84.

Results

5. For figures in general, it could be helpful to include a label of the cells applied (HL-60 vs. RF/6A) or a label of the bacterial species (e.g. Figures 7 A+B vs. C+D) to see this information quickly without searching through the legend.

RESPONSE: We have made these recommended changes to each of the figures that include multiple cell types and/or multiple bacteria.

6. Lines 100-113: This part does not report results but is rather an introductory paragraph on *A. phagocytophilum* and should thus be moved to the introduction section.

RESPONSE: We respectfully elect to keep this material (lines 107-119) at the beginning of the results section. Since the paper describes results with four different pathogens, we submit that providing a brief introduction to each pathogen at the appropriate Results section is the best means for framing the driving rationale each section. To include all of the background information for *A. phagocytophilum* as well as *C. burnetii* and the chlamydial species would make the introduction too cumbersome and diffuse. As quoted by Reviewer 3 above, "The results section frequently contains background explanations that could be transferred to the introduction but due to the utilization of four different models are helpful to understand the experiments and data at their place.", it appears that the reviewer sympathizes with this approach.

7. Lines 114 + 119 state that "Promyelocytic HL-60 and RF/6A endothelial cells" were used in this part but the figure 1 legend only names "HL-60 cells ... or human peripheral blood neutrophils". It appears that the RF/6A cell data are shown in S1 but also some important HL-60 data. There is continuous switching between these two figures. It would be easier to follow if all relevant data (figure 1 and S1) were combined into one larger figure with a common legend. The amount of details of the two legends is not consistent (also for later figures).

RESPONSE: As recommended, the data originally presented in Figures 1 and S1 have been solely reorganized into Figure 1. The cognate Results section (lines 106-154) and Figure 1 legend (lines 967-991) have been updated accordingly.

8. Figure 1: For faster grasping the information, it could be worthwhile to leave the first bar (DMSO) in Figure 1B empty (instead of full black color) to have this comparable to all other DMSO conditions in Figure 1.

RESPONSE: We agree. The requested change has been made.

9. Figure 2: The legend states, cells were fixed at 20, 24, 28 and 32 h in A to C, but figure A shows no 32h data - what is the reason?

RESPONSE: Thank you for pointing this out. Representative IFA images for 32 h are now included in Figure 2A.

10. Labeling of figures 2D-E-F could be improved and shortened by showing the times below the identical time points (20h etc.) and for example +/- for desipramine maintenance /removal of full and empty circle symbols with a legend for desipramine maintained or removed €, similarly for D (empty vs. full symbols) and E. It would be better to have D-E-F aligned in a column (for example swap figures C and D). The legend describes (lines 959-960) the maintenance (D) and removal (E) of desipramine but not (F) with later addition of desipramine.

RESPONSE: Times below the identical time points have been added. Panels D, E, and F are now aligned. The original legend did describe whether desipramine treatment was maintained throughout, initiated at 20 h, or removed at 20 h. Please see lines 993-997, which read as follows. "Desipramine was added to RF/6A cells prior to infection with *A. phagocytophilum* and treatment was either maintained throughout the time course (A, B, and D) or removed at 20 h (B and E). In some cases, desipramine was added to *A. phagocytophilum*-infected RF/6A cells beginning at 20 h (C and F)."

11. Lines 178-180: "Vimentin associated with all ApVs observed under both conditions (Figure 2A), indicating that this early ApV biogenesis event is not dependent on ASM activity." How do the authors prove that vimentin (despite its early recruitment and irreversible association with ApV lines described in lines 169-170) is associated with all (?) ApVs?

RESPONSE: We have verified this to be the case by performing an independent experiment in triplicate. DMSO- or desipramine-treated RF/6A cells were infected with *A. phagocytophilum*. The cells were examined by immunofluorescence microscopy and the percentage of *A. phagocytophilum*-occupied vacuoles (delineated by antibody labeling of intravacuolar bacteria with an antibody specific for the bacterial outer membrane protein, P44) with which vimentin associated was determined. Vimentin associated with 100% of ApVs for DMSO- and desipramine-treated cells for all three experiments. This data is now included in the Results (see lines 183-185) and Figure S1A (see lines 1091-1099).

12. Line 284 "Beginning on day 3, the first day on which there was any detectable increase in fluorescence signal..." - In figure 6, to which conditions are the time points compared (indicated by asterisks)? It is not obvious that there is already a clear increase at day 3 labelled "****" compared to day 0. Explain abbreviation Cb (*C. burnetii*) in the legend.

RESPONSE: For Figure 6A, to which we believe Reviewer 3 is specifically referring, statistically significant differences in fluorescence units in the absence (blue plot) and presence (red plot) of desipramine at each time point are indicated by ***. Regarding day 3, while the mean-fold difference between the two conditions is not large the difference is statistically significant because the respective triplicate values for each are very close. We double-checked to confirm this. The abbreviation, Cb, is now defined in the legend (please see line 1063).

13. Line 376 explain abbreviation NSF

RESPONSE: N-ethylmaleimide-sensitive factor. Since this is the only use we now state as N-ethylmaleimide-sensitive factor instead of NSF. See line 377.

Methods:

14. Lines 501f: To allow replication of presented results, used commercial antibodies should be listed with their product numbers not just the company, since many companies offer different antibodies for the same target.

RESPONSE: We agree. This will not only improve clarity, but also optimize reproducibility. All antibody product numbers are now listed (please see lines 500-522).

15. Lines 538f: For how long were the cells left with the FIASMA: for the entire duration of the infection or just the 1h prior to infection? The exchange of media (line 567f) suggests the latter but it's not clear from the sentence.

RESPONSE: The sentence in question has been revised to reflect that unless otherwise noted *A. phagocytophilum* infection continued in the presence of FIASMA. See lines 541-545.

16. Line 550f: "Infection conditions were optimized for both cell type and vessel for less than one internalized bacterium per cell." Is this true only for the infections with *A. phagocytophilum* described above or only for the

infections described below the sentence of for both? If for all, consider adding "all" and moving the sentence.

RESPONSE: We agree that this section could have been clearer and appreciate the suggestion. We have addressed this issue two ways. First, we removed "all" from the final sentence in that section. Second, we delineated descriptions of cultivation conditions for *A. phagocytophilum* infected, *C. burnetii* infected, and chlamydial infected cells as separate paragraphs. See lines 540-574.

17. Line 576: "0.5% Triton X-100" in PBS as given for PFA?

RESPONSE: This information is now written as "0.5% (vol/vol) Triton X-100 in PBS". See line 580.

18. Line 584: "0.1% saponin in 1% BSA" in PBS?

RESPONSE: This information is now written as "PBS containing 0.1% (vol/vol) saponin and 1% BSA". See line 588.

19. Line 601: "2.5% PFA" also in PBS as before?

RESPONSE: This information is now written as "2.5% (vol/vol) PFA in PBS". See line 587.

20. Line 602 "1% BSA" also in PBS?

RESPONSE: This information is now written as "1% (vol/vol) BSA in PBS". See lines 606.

21. Please always consistently state the exact solution or provide a general sentence (all in PBS).

RESPONSE: We have made such corrections throughout the methods.

22. Line 607 and later: "Hepes (Thermo Fisher Scientific)", please use HEPES buffer

RESPONSE: This has been corrected throughout.

23. Lines 622f: "Bacterial load was determined using primers specific for *A. phagocytophilum* 16S rDNA and host cell beta-actin using SsoFast EvaGreen Supermix (BioRad, Hercules, CA) as previously described (Ojogun et al., 2012)." Please provide exact primers used in this study. The reference Ojogun et al. 2012 lists primers for RT-qPCR including Ap 16S-527F and Ap 16S-753R that could be meant but none for (species specific to the used cells) beta-actin.

RESPONSE: Thank you for catching this citation error. The correct citation (Oki et al) has now been added. See line 627. This reference describes the sequences for the primers targeting β -actin and *A. phagocytophilum* 16S rDNA.

24. "Relative 16S rDNA was normalized to mouse beta-actin using the 2-DeltaDeltaCT 626 (Livak) method" - why mouse? Cell lines were not (all) from mice. Does this only refer to mouse blood samples?

RESPONSE: We apologize for the confusion. Thank you for pointing this out. We improved clarity two ways. First, we removed "all" from the sentence in question. Second, we clarified that the same were primers used to amplify β -actin from DNA isolated from HL-60 cells, RF/6A cells, and mouse blood because they target sequences that are conserved among human, primate, and murine β -actin, respectively (see lines 627-628).

25. Was RNA quality checked?

RESPONSE: RNA was confirmed to be genomic DNA-free by performing -RT reactions. RNA quality was checked via spectroscopy to verify that the 260:280 ratio was between 1.8 and 2.0. This is standard practice in our laboratory and RNA of this quality has always been sufficient for cDNA conversion and subsequent quantitative PCR.

26. It would be helpful to provide approximate amounts of template/dilutions used for these reactions (qPCR, cDNA synthesis, RT-qPCR). In general, MIQE guidelines should be followed where possible.

RESPONSE: This information has now been provided. Please see lines 629, 635-636, and 661-663.

27. Lines 657f: "qPCR as described above. Thermal cycling conditions used were 98°C for 2 min, followed by 40 cycles of 98°C for 5 s and 60°C for 30 s" Why was the thermal cycling protocol different from the one described above? Which beta-actin primers have been used for the mouse studies?

RESPONSE: The differences between the two thermal cycling protocols are that the annealing/extension step for amplifying DNA recovered from tissue culture cells occurs at 55°C for 10 s, but the conditions for DNA recovered from mouse blood require 60°C for 30 s. We empirically determined the 60°C for 30 s step to work best for DNA isolated from mouse blood as 55°C for 10 s simply did not sufficiently amplify the DNA. As stated above to Reviewer 3's previous inquiry, the β -actin primers used in this study target sequences that are conserved among human, primate, and murine β -actin.

Language

Please check spelling and correct notation particularly of chemicals throughout the text, for example:

28. Line 517f: Spelling mistakes:
Hoeschst 33342 should be Hoechst

RESPONSE: Thank you for catching this spelling error, which has now been corrected throughout.

29. amitryptiline (MilliporeSigma) should be amitriptyline

RESPONSE: Thank you for catching this spelling error, which has now been corrected throughout.

30. nortryptiline (MilliporeSigma) should be nortriptyline

RESPONSE: Thank you for catching this spelling error, which has now been corrected throughout.

31. Ca074-Me (MilliporeSigma) should be CA-074 Me

RESPONSE: Thank you for noting this error, which has now been corrected throughout.

32. Bodipy (e.g. figures 4D, E) should be consistently BODIPY

RESPONSE: Thank you for noting this error, which has now been corrected in Figure 4 panels D and E.

March 7, 2019

RE: Life Science Alliance Manuscript #LSA-2018-00292-TR

Dr. Jason Carlyon
Virginia Commonwealth University
Box 980678
Richmond, Virginia 23298-0678

Dear Dr. Carlyon,

Thank you for submitting your revised manuscript entitled "Functional inhibition of acid sphingomyelinase disrupts infection by intracellular bacteria". I appreciate the introduced changes and would be happy to publish your paper in Life Science Alliance pending final minor revisions:

- I would recommend to include image examples for the experiment quantified in Fig S1A
- Please add a scale bar in Fig6G
- We adhere to ICMJE author contribution guidelines (<http://www.icmje.org/recommendations/browse/roles-and-responsibilities/defining-the-role-of-authors-and-contributors.html>), please check whether these are met for all co-authors

A. FINAL FILES:

B. MANUSCRIPT ORGANIZATION AND FORMATTING:

Sincerely,

March 13, 2019

RE: Life Science Alliance Manuscript #LSA-2018-00292-TRR

Dr. Jason Carlyon
Virginia Commonwealth University
Box 980678
Richmond, Virginia 23298-0678

Dear Dr. Carlyon,

Thank you for submitting your Research Article entitled "Functional inhibition of acid sphingomyelinase disrupts infection by intracellular bacteria". It is a pleasure to let you know that your manuscript is now accepted for publication in Life Science Alliance. Congratulations on this interesting work.

DISTRIBUTION OF MATERIALS:

Again, congratulations on a very nice paper. I hope you found the review process to be constructive and are pleased with how the manuscript was handled editorially. We look forward to future exciting submissions from your lab.

Sincerely,
